# Global hotspots and correlates of emerging zoonotic diseases

Toph Allen[1], Kris A. Murray[2,3], Carlos Zambrana-Torrelio [ID] [1], Stephen S. Morse[4], Carlo Rondinini[5], Moreno Di Marco[6,7], Nathan Breit[1], Kevin J. Olival[1] & Peter Daszak[1]

Zoonoses originating from wildlife represent a significant threat to global health, security and economic growth, and combatting their emergence is a public health priority. However, our understanding of the mechanisms underlying their emergence remains rudimentary. Here we update a global database of emerging infectious disease (EID) events, create a novel measure of reporting effort, and fit boosted regression tree models to analyze the demographic, environmental and biological correlates of their occurrence. After accounting for reporting effort, we show that zoonotic EID risk is elevated in forested tropical regions experiencing land-use changes and where wildlife biodiversity (mammal species richness) is high. We present a new global hotspot map of spatial variation in our zoonotic EID risk index, and partial dependence plots illustrating relationships between events and predictors. Our results may help to improve surveillance and long-term EID monitoring programs, and design field experiments to test underlying mechanisms of zoonotic disease emergence.

[1] EcoHealth Alliance, 460 West 34th Street, 17th Floor, New York, NY 10001, USA. [2] Department of Infectious Disease Epidemiology, School of Public Health, Imperial College London, St Mary's Campus, Norfolk Place, London W2 1PG, UK. [3] Grantham Institute – Climate Change and the Environment, Imperial College London, Exhibition Road, London SW7 2AZ, UK. [4] Mailman School of Public Health, Columbia University, 722 West 168th St #1504, New York, NY 10032, USA. [5] Global Mammal Assessment Program, Department of Biology and Biotechnologies, Sapienza University of Rome, Viale dell'Università 32, 00185 Rome, Italy. [6] ARC Centre of Excellence for Environmental Decisions, Centre for Biosiversity and Conservation Science, University of Queensland, St Lucia, QLD 4072, Australia. [7] School of Earth and Environmental Sciences, The University of Queensland, St Lucia, QLD 4072, Australia. Correspondence and requests for materials should be addressed to P.D. (email: daszak@ecohealthalliance.org)

Emerging infectious diseases (EIDs) are a significant and growing threat to global health, global economy and global security[1, 2]. Analyses of their trends suggest that their frequency and economic impact are on the rise[3, 4], yet our understanding of the causes of disease emergence is incomplete. The majority of EIDs (and almost all recent pandemics) originate in animals, mostly wildlife, and their emergence often involves dynamic interactions among populations of wildlife, livestock, and people within rapidly changing environments[5–7]. The mechanisms underlying this process are likely complex, and occur in contexts that are often characterized by a paucity of systematically collected data[8].

Global efforts to reduce the impacts of emerging diseases are largely focused on post-emergence outbreak control, quarantine, drug, and vaccine development[3]. However, delays in detection of or response to newly emerged pathogens, combined with increased global urbanization and connectivity, have resulted in recent EIDs causing extensive mortality across cultural, political, and national boundaries (e.g., HIV), and disproportionately high economic damages (e.g., SARS, H1N1). Efforts to identify the origins and causes of disease emergence at local scales, and regions from which novel diseases may be more likely to emerge, are valuable for focusing surveillance, prevention, and control programs earlier in the chain of emergence, containing EIDs closer to their source, and more effectively limiting their subsequent spread and socioeconomic impacts[8].

A previous analysis of global EID trends modeled the spatial variation of "EID events", representing records of the first appearance of a pathogen in a human population related to increased distribution (e.g., new geographic location, new host species), incidence, virulence, or other factors[4]. The EID events were divided into four groups, including wildlife origin zoonoses[4]. To model the potential risk of disease emergence, these four groups were regressed as a function of human population density and growth, latitude, rainfall, and wildlife species richness. The results suggest that wildlife origin EIDs are more likely to occur in regions with higher human population density and greater wildlife diversity (mammal species richness)[8]. However, the study is limited in its mechanistic inference due, in part, to the lack of specificity of the predictors. For example, the effect of population density could represent anthropogenic environmental changes (human pressure on landscapes), human-animal contact rates, reporting biases, or a combination of these. Furthermore, a range of potential mechanisms may not be adequately represented by this predictor set; a lack of an effect of rainfall, for example, does not discount the potential for other climatic factors to play a role, and a lack of an effect of latitude could mean that it is simply a poor proxy for other more meaningful factors that nevertheless exhibit some latitudinal variation (e.g., temperature, habitat types, biodiversity, and GDP). Improving the predictor set to better target underlying mechanisms could improve model performance and our ability to explain spatial variation in EID risk.

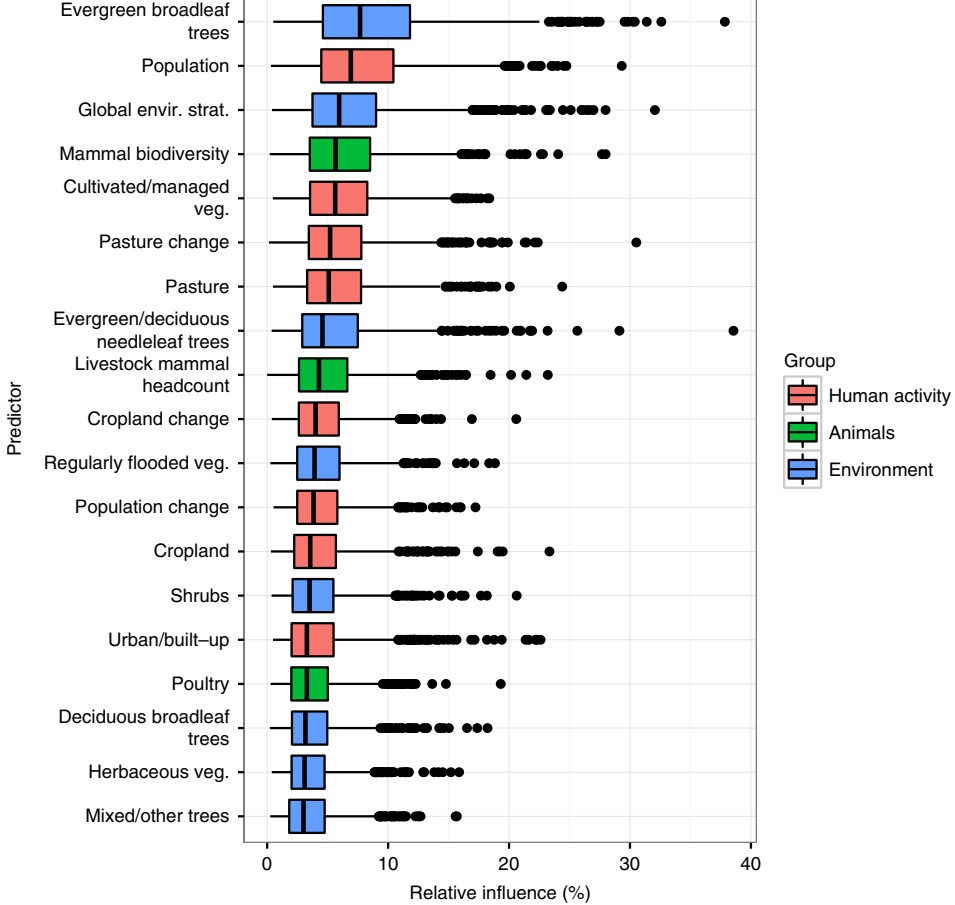

**Fig. 1** The relative influence of predictors on EID event occurrence probability. The box plots show the spread of relative influence across 1000 replicate model runs to account for uncertainty in EID event location (see above). Whiskers represent the minimum or maximum datum up to 1.5 times the inter-quartile range beyond the lower or upper quartile. BRTs do not provide *p*-values or coefficients, but rank variables by their relative influence in explaining variation in the outcome[26]

The current study aims to better analyze the mechanistic underpinnings of disease emergence for zoonotic EIDs of wildlife origin, while addressing some methodological limitations of Jones et al.[4] We focus on EIDs of wildlife origin, which are responsible for nearly all recent pandemics (e.g., Ebola, MERS), constitute the majority of the high impact EIDs from the last few

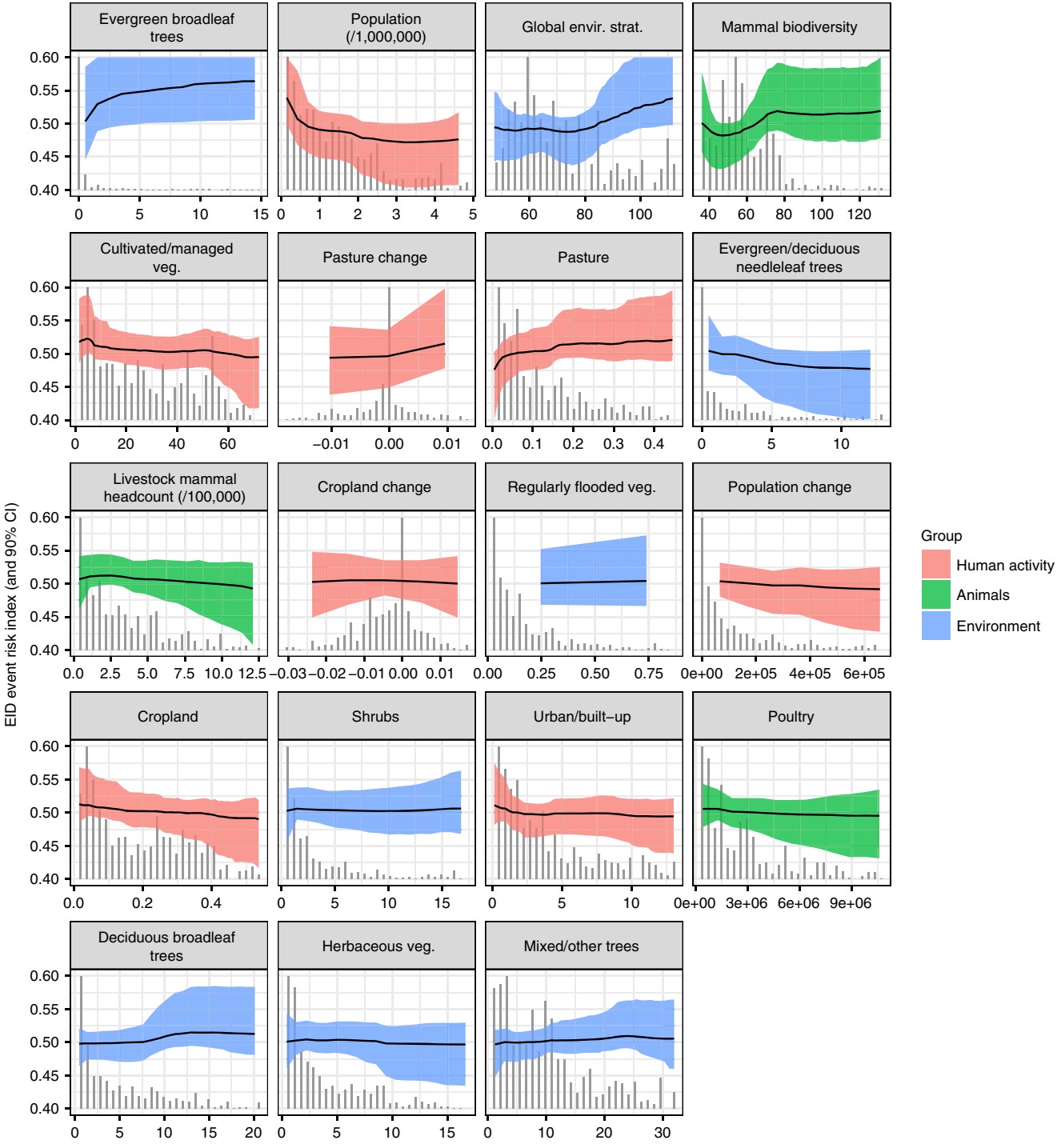

**Fig. 2** Partial dependence plots showing the influence on zoonotic EID events for all predictors in the weighted boosted regression tree model, ordered by relative influence. *X* axes show the range from the 10th to 90th percentiles of sampled values of predictors (e.g., number of mammal species per grid square for mammalian richness, or proportion of grid cell for a land cover type). Gray bars show histograms of predictor distribution along *X* axes. *Y* axes show the effect on the EID event risk index from that variable. Black lines show the median and colored areas show the 90% confidence intervals, computed using a bootstrap resampling regime incorporating uncertainty in EID event locations. The overall prevalence of our outcome, which indexes EID event risk, is fixed by the resampling regime between 0 and 1, with a mean at 0.5. *Y* axes are centered around the mean and scaled to 0.1 above and below. Partial dependence plots display the response for an individual variable in the model while holding all other variables constant[26, 61]. They allow a visualization of what are mostly non-linear relationships between drivers and the EID event risk index (in this case, after reporting effort is factored out). See Supplementary Note 3 for results of the model unweighted by reporting effort

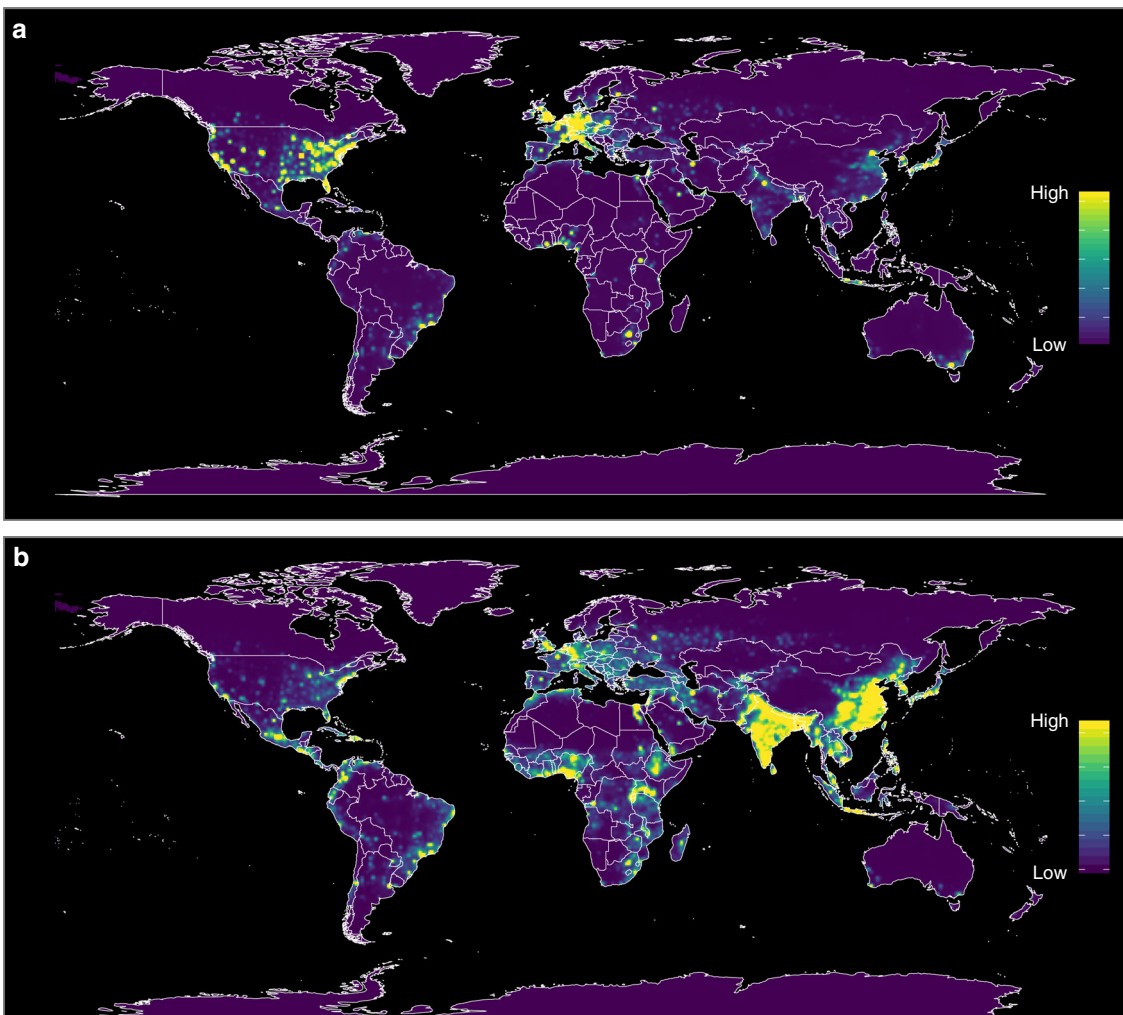

**Fig. 3** Heat maps of predicted relative risk distribution of zoonotic EID events. **a** shows the predicted distribution of new events being observed (weighted model output with current reporting effort); **b** shows the estimated risk of event locations after factoring out reporting bias (weighted model output reweighted by population). See Fig. 4 for raw weighted model output. Maps were created using standard deviation scaling, with the color palette scaled to 2.5 s.d. above and below the mean

decades, and are a significantly growing proportion of all EIDs combined[4]. We updated the EID database from[4], and employed a new modeling framework (boosted regression trees, BRT) to capture high-dimensional interactions and generate response functions for individual variables. We selected a refined set of spatial predictors for their relevance to a priori hypotheses on plausible mechanisms underlying zoonotic EID emergence, including proxies for human activity, environmental factors, and the zoonotic pathogen pool from which novel diseases could emerge, all key features of conceptual models of zoonotic spillover[7–11]. We used an improved data set of mammal species distributions[12], and included numerous data sets on measures of land use, land-use change and land cover. Furthermore, all data sets with sufficient temporal coverage were matched to events in the EID database by decade, such that covariates more accurately reflect the prevailing conditions at the time of disease emergence. We also constructed a novel proxy of reporting effort to match the spatial resolution of the other predictors, where previous studies have relied on coarse, country-level measures, and compared EID risk predictions with and without corrections for reporting effort. Finally, we accounted for spatial uncertainty in EID event data by random resampling to explicitly take into

account the difficulties of accurately geocoding EID events. Our results suggest that EID events are best predicted by the distribution of tropical forested regions, higher mammalian species richness, and variables relating to shifts in agricultural land use; and appear to occur more often in tropical regions. We identify specific areas and approaches where a research focus may identify more specific trends not apparent in our data.

## Results

**Variables in boosted regression tree models**. After factoring out reporting effort (in the weighted model), evergreen broadleaf trees (median 7.6% of the model's predictive power), human population density (6.9%), Global Environmental Stratification (climate) (5.9%), and mammal species richness (an aspect of biodiversity) (5.6%) had the largest relative influence over the distribution of EID events (Fig. 1). Across 1000 iterations of the model, no variables consistently emerged as much stronger predictors than others but an average ranking of predictor importance could be derived. Of the top predictors, evergreen broadleaf trees (representing tropical rainforests) exhibited an overall positive trend, human population density an overall negative trend, the Global Environmental Stratification (climate)

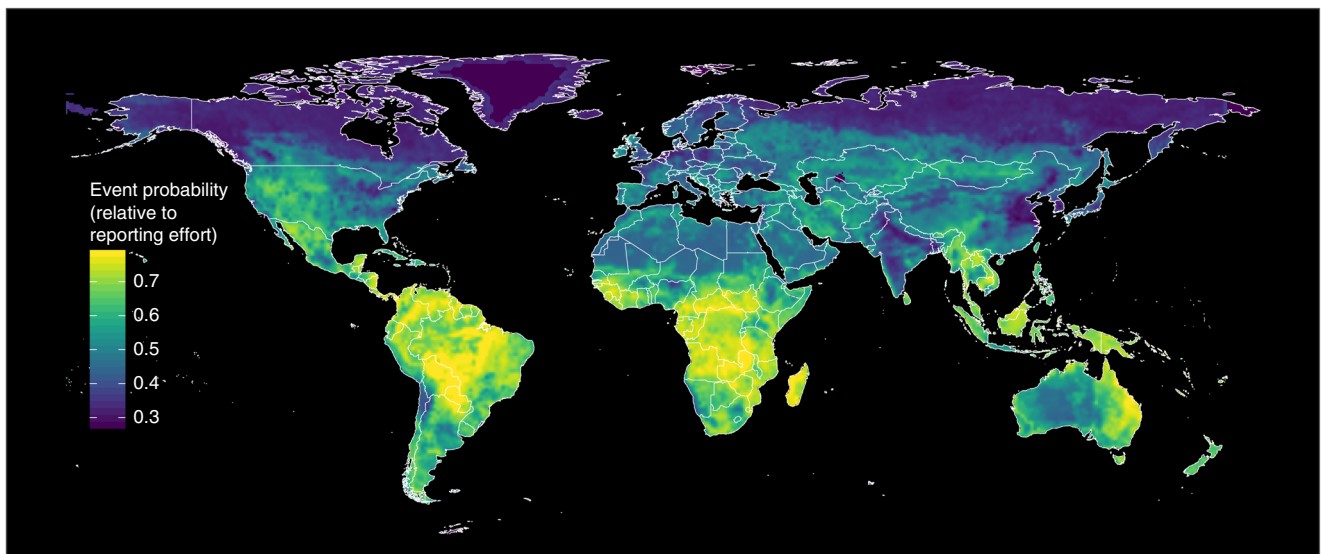

**Fig. 4** Heat map of weighted model response, i.e., EID risk relative to reporting effort. Value indicates the binomial probability that a grid cell sampled at that location will contain an EID event as opposed to a background sample, when drawing equal numbers of absence and background samples weighted by reporting effort (see Methods section). This layer was weighted by reporting effort to produce the "observed" EID risk index map (Fig. 3a) and by population to produce the risk index map with bias factored out (Fig. 3b)

an idiosyncratic trend towards warmer and wetter (i.e., more tropical) climates, and mammal species richness showed an idiosyncratic trend, with higher risk values at lower and particularly higher richness values (Fig. 2). After mammal species richness, three variables involving agricultural practices followed in importance: cultivated/managed vegetation (5.6%), pasture change (5.2%), and areas dedicated to pasture (5.1%). In the unweighted model, which did not account for reporting effort (Supplementary Note 3), urban/built-up land was by far the strongest predictor of observed events, explaining a median of 30.6% of the model's variation and exhibiting a distinct positive trend.

**Global distribution of EID risk index**. Relative to the observed risk index for EID events, the model's estimated risk index correcting for reporting bias (Fig. 3) is more concentrated in tropical regions. Areas of higher suitability for EID occurrence are fairly evenly distributed across the continents, with no major land mass free from areas predicted to be suitable for EIDs. In particular, areas of high population outside the tropics, such as cities in Europe, the United States, Asia and Latin America remain among areas at the high end of the risk index. Tropical regions in North America, Asia, Central Africa, and regions of South America have more extensive areas of predicted EID occurrence.

**Model performance and validation statistics**. Our model validation statistics were computed both for the weighted model —with a background, or absence, sample weighted by reporting effort, effectively computing statistics on the residuals of that variable—and our unweighted model, using a background sample uniform across land area. The weighted bootstrap model reported a median of 31.6% of deviance explained across the 1000 replicate models (empirical 90% confidence interval (CI) 15.9% to 50.5%), whereas the unweighted model explained a median 50.2% of deviance (empirical 90% CI 35.8% to 67.2%). Our weighted model's cross-validation statistics, computed over 100 runs of 10-fold cross-validation, varied depending on the weighting of the null validation sample. With validation absences weighted by reporting effort, the weighted model had a median AUC of 0.64,

with an empirical 90% confidence interval ranging from 0.54 to 0.69 (out of possible values between 0 and 1, with 0.5 indicating performance no better than random). The median True Skill Statistic (TSS) was 0.23 with an empirical 90% CI of 0.14 to 0.33 (out of a range of −1 to 1). These indicate low to moderate predictive performance[13–15]. Evaluated against an unweighted null, the weighted model had a median AUC of 0.78 (90% CI (0.75, 0.81)) and a median TSS of 0.43 (90% CI (0.37, 0.50)). The unweighted model evaluated against to an unweighted null, had a median AUC of 0.77 (90% CI (0.73, 0.81)) and a median TSS of 0.44 (90% CI (0.37, 0.50)).

## Discussion

We developed a spatial model to describe the global spatial patterns of zoonotic EIDs. Our main model (the "weighted model") factored out clear effects of reporting effort, which otherwise biases our ability to interpret EID event observations. It ranked risk factors according to their predictive power, capturing both their main effects and potential interactions with other variables, and we derived the directionality and shape of their relationships to EID events for graphical interpretation. Our results suggest that the risk of disease emergence is elevated in tropical forest regions, high in mammal biodiversity, and experiencing anthropogenic land use changes related to agricultural practices[16–18].

The link between mammal biodiversity and zoonotic disease emergence has been identified previously[4] and hypothesized widely[8, 19]. Areas with tropical forest and high mammalian biodiversity were elevated on our EID risk index (henceforth "EID risk"), although the uncertainty of the estimates was high. It may be that these variables represent the same mechanism, as tropical forests are generally areas of high biodiversity[20], and the apparent association may be attenuated by the presence of both in the model. This trend is consistent with existing hypotheses, which suggest greater host biodiversity, increases the "depth" of the pathogen pool from which novel pathogens may emerge, which in turn increases the potential for novel zoonotic pathogens to emerge[21]. There is a large literature on the relationship between biodiversity and infectious disease risk in people, with some studies suggesting that high host

biodiversity decreases risk or that biodiversity loss may increase risk (i.e., the dilution effect)[22], while others refute the generalizability of this[23, 24] or suggest disease richness or prevalence increases with increasing wildlife species richness[13]. Our findings look at the global scale and a large group of pathogens, and so do not speak directly to this debate: although the dominant trend is an increase in risk of disease emergence with higher mammalian richness, this neither rules out nor substantiates the possibility of a dilution effect for specific diseases. Rather, it is consistent with previous suggestions that the relationship between biodiversity and disease risk is complex, context-specific and idiosyncratic[23].

When not accounting for reporting effort (unweighted), our model showed urban land as having a very strong positive association with EID events. However, this can be interpreted as an effect of reporting bias, since (1) urban land was also strongly associated with our measure of reporting effort, and (2) fitting our weighted model, relative to reporting effort, attenuated this effect. Similarly, although population density was not found to be an important predictor in the unweighted model (median relative influence 2.2%), weighting the model by reporting effort drove up its importance (median rel. inf. 6.9%), such that EID risk was inversely related to population density. Population density was also included in the reporting effort model, but was not as strong a predictor (rel. inf. 3.6%) as urban land (rel. inf. 45.2%). Theoretically, population has a baseline multiplicative effect on human disease events[25]—of which EID events are a subclass—and their detection is modulated by reporting effort. Reporting effort appears to be associated with urbanization, but reporting effort and urbanization are also both products of human population. We did not attempt to fully disentangle these factors, instead using our measure of reporting effort to present a map of emerging infectious disease hotspots with bias "factored out" (described below in Methods section).

Our reporting effort measure was created by matching place names in a subset of the biomedical literature. The BRT model of reporting effort model suggested that the distribution of this effort was strongly and positively related to urban areas. This could be because our extraction of place names biases the outcome toward urban areas, or it may accurately represent the true distribution of reporting toward urban areas, or a combination of the two. In either case, our reporting effort data set is likely to be a large improvement over similar previous studies that have used country-level data to control heterogeneous reporting effort in better-than country-level spatial analyses of disease risk[4, 25] (detailed fully in Supplementary Methods).

The work presented here builds on previous research[4] in a number of important ways to advance our understanding of wildlife origin zoonotic disease emergence. First, our model building approach explores the explanatory value of a large collection of globally gridded data on environmental, demographic, and host diversity variables, including newly developed models of mammal distributions and richness patterns. This has allowed us to close the gap between predictors and a priori mechanistic hypotheses specifically relevant to zoonotic disease emergence from wildlife reservoirs. Second, we adopted a machine-learning modeling approach (boosted regression trees) suited to the analysis of complex ecological data[26], and used various resampling regimes to measure and visualize multiple sources of uncertainty (model uncertainty, spatial uncertainty of EID events, and temporal uncertainty of covariates matching with events) and predictive performance. Third, we have attempted to improve how the model accounts for uneven global distribution of surveillance and research on disease event detection (i.e., report effort). This includes an algorithm-based approach to more

realistically map reporting effort and shows the significant implications that a finer-scale, sub-national resolution variable for reporting effort can have for a model. Finally, we were able to temporally match predictors to events.

Despite using a more flexible modeling framework, there are limitations to our approach. When differentiating between EID events and a uniformly weighted background sample, our weighted and unweighted models had an AUC of 0.78 and 0.77, and a TSS of 0.43 and 0.41, respectively, indicating moderate predictive performance. However, against a background sample weighted by reporting effort, our weighted model had an AUC of 0.61 and a TSS of 0.18, indicating low–moderate performance. These statistics indicate much unexplained variation. While broad changes in zoonotic EID relative risk are evident in the partial dependence plots, in areas of elevated risk CIs are generally wide enough that quantitative relationships remain uncertain.

Wherever possible, we tried to define and incorporate uncertainty into our model (e.g., correcting for uncertainty in location by sampling EID events from within known areas of occurrence, and correcting for literature-level biases by weighting background samples by our measure of observation effort). Multiple factors contribute to this uncertainty. First, analyses were conducted using gridded data at 1° WGS84 resolution (c. 100 km at the equator), the same resolution used previously[4]. Our choice of resolution for predictor data sets was constrained by data availability, since all were downscaled to the lowest common spatial resolution. Second, CIs are widest in regions for each variable where fewer grid cells were sampled. Since our weighted model sampled fewer grid cells proportional with reporting effort, these represent areas where more reporting effort —including ground-truthing studies—may increase confidence. Third, another limitation shared with ref. [4] is the underlying accuracy and suitability of EID event data, which were drawn from a review of published literature. Individual studies carry their own biases, inaccuracies, and different approaches to collecting and documenting data, and this alone adds an unknown amount of imprecision and potential bias to our outcome data set. Finally, our goal of creating a single model, to look for common trends in emerging wildlife origin zoonotic diseases, likely imposes limitations on the specificity of trends we can examine. In reality, different classes of diseases (e.g., viruses versus bacteria) and indeed individual diseases have their own unique biology and ecology, with different drivers and sets of conditions being more or less important in shaping the emergence process[27]. Because of these limitations, we refrain from making specific (e.g., city by city) interpretations of the model's output, rather noting broad trends in geographic regions and environment types of intererest.

Wide confidence intervals in areas of elevated EID risk suggest areas for future study, and underscore the need for targeted long-term disease surveillance and monitoring in these areas. Collection of more accurate spatiotemporal data on events surrounding disease emergence, including initial emergence events, using a combination of large scale field research (e.g., USAID's PREDICT project[28]) and digital disease detection tools[29] would help alleviate this issue in the future by generating more consistent data on a larger scale, potentially automatically[30]. These data sets will aid efforts to better define the point at which a disease becomes "emerging", and allow the programmatic definition and examination of different definitions of emergence (e.g., first appearance vs. increasing incidence, etc.) in testable form[31].

Future work may be able to enhance the predictive power of this approach by focusing on even tighter classes of disease, taxonomic groups of pathogens and hosts, or transmission modes, and building models to forecast changes in risk

**Table 1 List of predictor layers included in the model**

| Variable | Unit per grid cell | Type | Source data set | Processing | Temporal resolution |
|---|---|---|---|---|---|
| Human population | Population | Human activity | GRUMP | Rescaled | Decadal |
| Population change | Change in population | Human activity | GRUMP (calculated) | Calculated from rescaled layers | Decadal |
| Cropland | Proportion | Human activity | HYDE | Rescaled | Decadal |
| Cropland change | Change in proportion | Human activity | HYDE (calculated) | Calculated from rescaled layers | Decadal |
| Pasture | Proportion | Human activity | HYDE | Rescaled | Decadal |
| Pasture change | Change in proportion | Human activity | HYDE (calculated) | Calculated from rescaled layers | Decadal |
| Urban land | Percentage | Human activity | EarthEnv | Rescaled | Decadal |
| Managed/cultivated vegetation | Percentage | Human activity | EarthEnv | Rescaled | Static |
| Mammalian species richness | Count of species | Animals/hosts | Global Mammal Assessment | Reprojected, rescaled | Static |
| Domestic mammal headcount | Count of animals | Animals/hosts | GLW | Rescaled, summed buffalo, cattle, goat, pig, sheep headcounts | Static |
| Poultry headcount | Count of animals | Animals/hosts | GLW | Rescaled | Static |
| Global environmental stratification | Global environmental stratification | Environment | GEnS | Rescaled | Static |
| Evergreen/deciduous needleleaf trees | Percentage | Environment | EarthEnv | Rescaled | Static |
| Evergreen broadleaf trees | Percentage | Environment | EarthEnv | Rescaled | Static |
| Deciduous broadleaf trees | Percentage | Environment | EarthEnv | Rescaled | Static |
| Mixed/other trees | Percentage | Environment | EarthEnv | Rescaled | Static |
| Shrubs | Percentage | Environment | EarthEnv | Rescaled | Static |
| Herbaceous vegetation | Percentage | Environment | EarthEnv | Rescaled | Static |
| Regularly flooded vegetation | Percentage | Environment | EarthEnv | Rescaled | Static |
| Reporting effort | Weighted number of mentions in publications | Observation bias | (Internal) | (See methods) | Static |

distribution or to examine more specific mechanistic hypotheses. For example, our model includes a single layer representing total mammal species richness, whereas recent work has shown that the number of zoonotic viruses varies across mammal species and taxa[32]. Efforts to examine the commonalities of disease emergence may benefit from incorporating host-specific or disease-specific models in a hierarchical approach, allowing certain parameters to vary across diseases, disease classes, or other properties.

Despite shortcomings, our improvements to the earlier model allowed us to find quantitative support for previously only hypothesized factors that increase the risk of EID events. Our findings, therefore, have broad implications for surveillance, monitoring, control, and research on emerging infectious diseases. Like Jones et al.,[4] we find that EID events are observed predominantly in developed countries, where surveillance is strongest, but that our predicted risk is higher in tropical, developing countries.

Our spatial mapping has direct relevance to ongoing surveillance and pathogen discovery efforts (http://www.globalviromeproject.org/). It shows that the global distribution of zoonotic EID risk (and the presence of EID "hotspots") is concentrated in tropical regions where wildlife biodiversity is high and land-use change is occurring. These regions are likely to be the most cost effective for surveillance programs targeting wildlife, livestock or people for novel zoonoses, and for pandemic prevention programs that build capacity and infrastructure to pre-empt and control outbreaks[28]. Further honing the EID risk index within regions and countries might also inform the planning of large land-use change programs such as logging and mining concessions, dam-building, and road development[33]. These activities carry an intrinsic risk of disease emergence by increasing human or livestock contact with wildlife in new regions or by disrupting disease dynamics in reservoir hosts[21, 34], and have been repeatedly linked to outbreaks of novel EIDs.

Similarly, the partial dependence plots allow a deeper understanding of the largely non-linear relationships between EID drivers and disease emergence that can be used to design field experiments to test specific and generalizable hypotheses on the drivers of zoonotic disease emergence. These should include field sites along land use gradients within EID hotspot countries where controlled sampling protocols are used to identify how wildlife biodiversity, known and unknown pathogen diversity (e.g., using viral family level degenerate primers for PCR[35]), and human contact with wildlife varies across a landscape. Such an approach will provide a way to identify the fine-scale rules that govern disease emergence and provide a richer understanding of what drives EID risk on-the-ground, a critical extension of this modeling approach.

## Methods

**Zoonotic EID events as response variable**. We followed the definition of an emerging infectious disease and an EID event used in ref. [4]—specifically, events documented in the scientific literature denoting the first emergence of pathogen in a human population where that pathogen was classified as "emerging" due to recent spillover from an animal reservoir, a significant increase in its incidence or geographic distribution in the human population, a marked change in its pathogenicity or virulence, or other factors. In this study we focus only on EID events of wildlife origin ("wildlife zoonoses") because these represent the majority of EID events in the most recent decade studied, are increasing significantly as a proportion of all EIDs after correcting for reporting bias, include most of the highest impact EIDs of recent decades (e.g., Ebola viruses, Nipah virus) and almost all recent pandemics (e.g., pandemic influenza viruses, SARS). Data on EID events were derived from an updated version of the database originally used by ref. [4] (Supplementary Data 1), which contained EID events ranging from 1940 to 2004 ($n = 335$ total, $n = 145$ for wildlife zoonoses (43.3% of all EIDs)). We updated the database to include EID events for wildlife zoonoses through 2008 ($n = 224$), following the methodology in ref. [4] so as to include only diseases reported in the peer-reviewed literature, where there is evidence that a disease is emerging for one of the reasons laid out above. In addition, we only included the first emergence of a new disease-causing agent, such that the MERS Coronavirus was included, but not reports of new strains of Ebola virus. For each EID event, data were derived from the literature, if available, for date, location (see below), pathogen genus and species, zoonotic origin and type, and associated or hypothesized drivers, following ref. [4]. Location data for initial EID emergence events were variable in their geographic specificity, ranging from precise coordinates to broader regions (e.g., municipalities, counties, districts) or entire continents depending on details reported in the primary literature. A spatial polygon was created for each event that represented the most precise municipal region the EID event was known to have occurred in. All EID event polygons, regardless of precision, were included in our bootstrap resampling framework; removing those with geographic uncertainty (e.g., those with only country-level resolution) may artificially inflate the apparent certainty of our model, and our resampling scheme limits their impact to appropriate levels. Events with precise coordinates were also assigned a polygon for consistency of data format, but rather than using a municipal boundary, the event was assigned a 5 km circular buffer zone. EID polygons were subsampled for model fitting as described below. Because our model matches EID events with decadal population and land use data (described below), we restricted our analyses to decades for which covariate data exist, excluding events before 1970 and leaving $n = 147$ records for analysis (66% of wildlife zoonosis events).

**Explanatory variables**. We compiled spatial data layers for 20 predictors in four broad categories to decompose which factors are associated with zoonotic disease emergence. These reflected the most frequently hypothesized drivers of zoonotic disease emergence and included (Table 1): human presence/activity, animals/hosts, the environment, and reporting effort. Explanatory variables came from a variety of data sources, and all were rescaled or transformed to a spatial grid of 1° resolution (WGS84, c. 110 km at the equator) prior to their use in models. Full details of sources, original resolutions and rescaling are presented in Tables 1 and 2.

**Table 2 Original resolutions and extents of source data sets**

| Source data set | Spatial resolution | Temporal resolution and extent |
|---|---|---|
| GRUMP (Global Rural–Urban Mapping Project)[39] | 0°5′ | 5 years, 1970–2000 |
| HYDE (History Database of the Global Environment)[43] | 0°5′ | 10 years, 1900–2000 |
| GMA (Global Mammal Assessment)[12] | 300 m | N/A |
| GLW (Gridded Livestock of the World)[48] | 0.05° | N/A |
| GEnS (Global Environmental Stratification)[53] | 0°0′30″ | N/A |
| EarthEnv[55] | 0°0′30″ | N/A |

"Human Activity" data were compiled and eight predictors derived based on the following rationale: (1) Population density likely influences EID risk in two discrete ways. First, as EID events are defined as diseases emerging in the human population, their frequency—before the effects of other predictors—is assumed to be proportional to population density, with the other predictors modifying the per-person risk of EID events. To represent this, we treated human population as a baseline multiplicative factor in our models[36]. Second, population density may affect transmission dynamics such that EID events in areas of denser population may be more likely to produce outbreaks large enough to be detected[37]. We used the Global Rural–Urban Mapping Project[38] human population data set, which provides gridded estimates of human population every five years for 1970–2000. (2) Population change acts as a proxy for changing demands on ecosystems leading to environmental perturbation, which has been hypothesized to drive disease emergence[21]. We created a measure for population change by calculating the inter-decadal difference of human population per grid cell. (3) Land-use type represents largely anthropogenic influence on the landscape (as opposed to 'land cover' below) and has been hypothesized to play a role in disease emergence and spatial distribution[19, 21, 39–41]. We used the HYDE data set which estimates the percentage of land-use types in each grid cell of a global data set every ten years for 1900–2000[42] to derive predictors representing percentage of land used for cropland and percentage used for pasture. We also include the layers for Urban Land and Managed/Cultivated Vegetation from the EarthEnv data set, described below under "Environment", in this category, as they index human impact on the environment. (4) Land-use change has been hypothesized as a key driver for disease emergence by perturbing ecosystems and bringing humans into close proximity with wildlife[5, 7, 8, 21, 27]. We created metrics of change for pasture and cropland by calculating the between-decade difference in values for each grid cell for cropland and pasture.

For data sets with multiple temporal layers (human population, cropland, and pasture), we included the intersection of available dates in different data sets (decades 1970–2000) and calculated inter-decadal change layers by differencing consecutive decades. All presence and absence samples drawn for each event (see below) were matched to the nearest decadal layers (years ending in 5 were rounded up) and the change layer for the decade they fell in.

"Animal/host" data were represented by two predictors: (1) Mammalian biodiversity. The diversity and prevalence in a host population of potentially zoonotic pathogens in an area is hypothesized to be a key factor in the risk of novel pathogen emergence[8, 21, 43]. However, spatial data on global pathogen diversity do not currently exist, and it is estimated that we have identified less than 1% of mammalian viral diversity[35]. Consistent with previous studies, we therefore assume that the number of available pathogens in an area is proportional to the diversity (species richness) of wildlife species[4, 5, 35, 44]. The overwhelming majority of emerging zoonoses have mammalian hosts[45], and global biogeographic patterns of human infectious diseases is highly correlated with global patterns of mammalian diversity[30]. We therefore used mammal biodiversity (species richness), measured as number of mammal species per grid cell as a proxy for pathogen species richness. To do this, we used the most up to date mammal species distribution maps available, derived from species distribution ranges filtered according to species-specific habitat preferences[12]. These habitat suitability models reflected species preferences for land cover types, their altitudinal limits, their tolerance to human presence, and their relationship with water bodies. The full-resolution mammal biodiversity data (representing all 5291 terrestrial mammal species)[12] was rescaled to the study grid by summing the number of species' distributions that overlapped each grid cell; (2) Domestic animal density. A number of past EID events with wildlife origin have emerged through farmed or domestic animal intermediate or amplifier hosts (e.g., Hendra and Nipah virus, SARS). In addition, there is growing evidence that the global trend of intensification of livestock production increases the emergence risk of novel wildlife origin zoonoses, e.g., Nipah virus in Malaysia[46], influenza viruses, and others[6]. We used the Gridded Livestock of the World (GLW) data set[47], which contains data for poultry, goat, buffalo, cattle, sheep, and pig headcounts. We summed mammals to a single predictor (livestock mammal headcount) and retained poultry as a discrete predictor.

We analyzed eight predictors from two data sets representing "Environmental" variables: (1) Climate. Climatic factors have been repeatedly hypothesized as important in the global biogeography of human infectious diseases, including EIDs[30, 48, 49]. Climate may influence disease distribution through enhanced

suitability for vectors of wildlife origin zoonoses (e.g., West Nile virus), more rapid vector reproduction rates and biting rates, changes in the efficiency or rates of pathogen transmission among hosts and vectors, and changes in the ability of pathogens to persist in the environment, among other factors[50, 51]. Climate was represented by a single layer in our study, the Global Environmental Stratification[52], which uses a quantitative model to stratify the Earth's surface into zones of similar climate on a single scalar measure, where higher values equate to warmer, wetter (more tropical) regions; (2) Land cover type: Land cover type is associated with the distribution of terrestrial mammals[12] and other taxa[53], potentially exposing humans present to different assemblages of viral species. It is also likely that the types of contact between wildlife and people vary with land cover type. For land cover, we used the EarthEnv data set[54], which divides the Earth's surface into 12 classes. These include different classes of natural ecosystems, urban land and cultivated vegetation (grouped with "Human Activity" above). We excluded barren areas, open water and snow/ice due to a lack of biologically plausible mechanisms for disease emergence. EarthEnv represents each class as a percentage per grid cell.

**Reporting effort.** The distribution of reported EID events is likely strongly influenced by an inconsistent spatial distribution of detection and reporting of disease outbreaks. Previous studies have used proxies of reporting effort such as the interpolated locations of known sampling sites ("sampling effort")[55]; frequency of countries of residence for all authors of all articles in the Journal of Infectious Disease ("reporting effort")[4]; and PubMed searches for keywords for each country ("reporting bias")[25]. Other studies have used occurrence records for a similar class of observations as a surrogate for background sampling effort; for example, in ecology, modeling the distribution of a particular species and utilizing occurrence records from multiple other species to represent background samples[56].

We adapted these approaches by deriving an index for reporting effort based on the spatial distribution of toponyms (place names) in peer-reviewed biomedical literature. We wrote a Python package, PubCrawler (see Supplementary Methods for full details), to search the full text of each of the 1,266,085 (as of April 2016) articles in the PubMed Central Open-Access Subset (PMCOAS)[57] for toponyms from the GeoNames database[58], which includes data on population (if appropriate), country, and geographical coordinates for each toponym. PubCrawler uses a set of heuristics, based on textual and geographic features of the identified toponyms, to minimize the number of false positives and select amongst ambiguous matches. We selected articles matching terms from the Human Disease Ontology[59] and exported extracted toponyms. After excluding a further round of potentially spurious matches, place name matches were assigned a weight, normalized by article, and then summed to the study grid. To impute missing data (resulting in a number of zero-value grid cells) and smooth noise in the raw output, we fit a Poisson boosted regression tree model (using human population, accessibility, urbanized land, DALY rates, health expenditure, and GDP as predictors), and used this to represent reporting effort in our model. This approach produced a layer that adequately represented the underlying data while achieving a similar coverage of grid cells to other layers.

**Statistical framework.** We used boosted regression trees (BRT) to model EID occurrence[26, 48, 60] and to determine how conditions varied between locations where EID events have been observed compared to areas where they have not. BRTs handle non-linear relationships and higher order interactions among many variables more robustly than many other modeling methods, and are robust to monotonic transformations of data[26, 60]. They fit potentially complex, non-linear relationships by aggregating the predictions of multiple simpler models, and are trained iteratively on random partitions of the data[26, 60]. In addition, predictive accuracy of BRTs, as determined by common validation methodologies (e.g., Area Under the Curve of the Receiver-Operator Characteristic (AUC of the ROC), True Skill Statistic (TSS)), frequently exceeds conventional linear methods[26]. Unlike conventional models, they do not produce confidence intervals or p-values.

**Resampling regimes.** We employed various resampling techniques to incorporate our measure of reporting effort[56, 61], estimate the predictive power of our models, account for spatial uncertainty in EID events[15], and generate empirical confidence intervals for effects representing both sampling uncertainty and spatial

uncertainty[62]. Each time an event was sampled, one presence point and one absence point were drawn (artificially fixing overall prevalence at 0.5)[15]. The presence point was from the grid cells overlapped by that event's polygon, and the absence point from all grid cells; both were weighted by reporting effort (the effect of weighting presence points by reporting effort made little difference for points with small, precisely specified occurrence polygons, and for events with high uncertainty it acted as a prior, specifying that, in the absence of other knowledge, the event was more likely detected where reporting effort was higher).

All replicate BRT models were fit using the R packages dismo and gbm[26]. The function gbm.step() was called with the parameters tree.complexity = 3 (governing interaction depth), learning.rate = 0.0035 (setting the "shrinkage" applied to individual trees), and n.trees = 35 (governing the initial number of trees fit, as well as the "step size" or number added at each step of the stagewise fitting process)[26]. These values were selected through an iterative process, starting with the default parameters, adding tree complexity, and tuning the shrinkage and step size parameters to achieve successful gradient descent consistently across resampling runs, following refs. [26, 62]. With the final parameters, the BRTs composing the bootstrap model fit a mean of 1005 trees.

Our main model used a bootstrap resampling regime, which was used to fit 1000 replicate models. For each model, 147 events were drawn randomly with replacement from the set 147 EID events of interest, and for each selected event, 1 presence and 1 absence value were drawn as described above. The fitted models were used to generate Relative Influence box plots and Partial Dependence plots with empirical 90% confidence intervals. The mean of the predictions of these models were used to generate all maps.

To compute validation statistics (described below), we conducted 100 rounds of 10-fold cross-validation[15, 62]. In each round, a single presence and absence sample were drawn for each event, which were assigned randomly to ten groups. Each group in turn was held out, and a model was trained on the remaining groups' samples. The model's predictions for the presence and absences samples of the held-out group were used to construct confusion matrices, and calculate the AUC and TSS. This process was repeated 100 times, and the median, 0.05 and 0.95 quantiles for all scores were reported.

**Factoring reporting bias out**. We assumed that the distribution of observed EID events was conditional on the distribution of reporting effort across the globe following[56]. We fit our main, "weighted" model with grid cells sampled relative to reporting effort. The model thus produced a response relative to reporting effort (Fig. 4). We multiplied this response by the value of reporting effort in each grid cell to map the index of observed EID event risk (Fig. 3a).

We produced the estimate of the risk index after factoring out reporting bias (Fig. 3b) as follows. We assumed that the optimal distribution of reporting effort for human disease events in a location is proportional to the distribution of the human population. In reality, other unmeasured factors likely affect this. However, given this assumption, we can define reporting bias as proportional to the ratio of reporting effort to the human population (Fig. 4).

$$\text{Reporting bias} \propto \frac{\text{Reporting effort}}{\text{Population}}$$

When bias is known, it is possible to estimate the true distribution of a phenomenon by "factoring bias out"[56]. In ecological studies, this generally means dividing by the measured "survey effort", assuming that the optimal distribution of search effort is uniform across the landscape.

$$\text{True risk index} \propto \frac{\text{Observed risk index}}{\text{Reporting bias}}$$

We posit that, in the case of human disease events, uniform search effort across a landscape is also suboptimal, and that it is safer to assume optimal reporting effort distribution would be proportional to the human population. In this case, we remove "bias" by factoring out measured reporting effort and factoring in assumed optimal effort, and obtain a hypothetical map of the true event risk index, thus:

$$\text{True risk index} \propto \text{Observed risk index} \times \frac{\text{Human population}}{\text{Reporting effort}}$$

**Model validation and performance**. We used multiple tools for model validation and performance. For our bootstrap model, we calculated deviance explained using the gbm.step() function[26] and also derived median and empirical 90% CIs by taking the 0.05, 0.5, and 0.95 quantiles of those values for the replicate models. Since this model is fit relative to reporting effort, percentage deviance explained is calculated relative to that variable. For the ten-fold cross-validation runs, we calculated the AUC, a threshold-independent measure of model predictive performance that is commonly used as a validation metric in species distribution modelling[63]. The AUC can be interpreted as "the probability that the model will rank a randomly chosen presence site higher than a randomly chosen absence site"[64], or more accurately in our application, a measure of a model's performance to discriminate EID events from random points[56]. Because the use of AUC has been criticized for its lack of sensitivity to absolute predicted probability and its inclusion of a priori untenable prediction thresholds[13], we also calculated the True Skill Statistic (TSS)[15].

Because all test statistics and figures from our main model are relative to the reporting effort measure, we also ran "unweighted" models. We expected these would score yield higher cross-validation scores, since we expected that reporting effort would be correlated both with some important predictor variables and the outcome, and weighting background samples uniformly rather than according to this variable would present a clearer contrast. To avoid bias from land area in the WGS84 grid cells, we additionally weighted our "unweighted models" by land area per grid cell. The figures from these models are presented fully in Supplementary Information.

**Code availability**. All data and code used to generate the models are available on GitHub (doi: 10.5281/zenodo.400978)[65], as is the code used to generate the reporting effort layer (doi: 10.5281/zenodo.400977)[66].

**Data availability**. The data sets analyzed during this study are included in this published article and its Supplementary Information Files, with the exception of EID Event shape files, which are available from the corresponding author on reasonable request.

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

## Acknowledgements

This work was made possible by the generous support of the American people through the United States Agency for International Development (USAID) Emerging Pandemic Threats PREDICT (Cooperative Agreement No. AID-OAA-A-14-00102). The natural language processing software described was sponsored by the Department of the Defense, Defense Threat Reduction Agency (Project No. J9CBA14212). The contents are the responsibility of the authors and do not necessarily reflect the views or the policy of USAID or the United States Government, and no official endorsement should be inferred. We thank Liam Brierly (Univ. of Edinburgh) for collating new EID data.

## Author contributions

T.A. and K.M. designed the statistical approach, with contributions from K.J.O. and C.Z.-T. The EID database was updated under P.D.'s supervision. T.A. wrote the modeling code and generated the figures, and N.B. and T.A. wrote the code to generate the publication bias layer. C.R. and M.D.M. contributed the mammal species richness data set. T.A., K.M., K.J.O., and P.D. wrote the manuscript, with all authors contributing edits.

## Additional information

**Competing interests:** The authors declare no competing financial interests.

