## [Peer Review file · Nature Communications]

Reviewers' comments:

Reviewer #1 (Remarks to the Author):

This is a straightforward report of a new interactive data base on emerging infectious diseases. As such it is not a typical example of primary research, but the database of potential importance to researchers and students.

There is little to criticize in this straightforward and brief description. I found it most helpful to go to the website and "play" with various parameters as I was reading the ms.

The title is a bit misleading as it does not indicate that the ms. describes website data and how to access and use it. Also, I must say, the actual writing is dry and not very interesting, whereas the website itself is quite interesting. Perhaps there is a way to put more pizzazz into the text.

The term "emergence" in their text is used to include "re-emergence" as well, and this should perhaps be explicitly spelled out.

One difficulty in assessing the power of the site data is that if anything has been missed how would one know it?

Also there must be some arbitrariness to the criteria for including a disease as newly emerging, and this is not very clear. For example, is every new salmonella type, or every new strain of an evolving virus a new disease. Was, for example, Bundibugyo ebolavirus disease an EID or not, and how is that decided? What about drug resistant viruses? Are they treated differently than drug resistant bacteria?

It is curious that they did not try to include past emergences, from before 1940. Hopefully that can be added at a later date.

Finally, I think readers will be skeptical about possible subjectivity in assigning disease drivers such as ecosystem change or war.

Reviewer #2 (Remarks to the Author):

In this article the authors outline a series of major advances to the methodology of Jones et al. 2008 and update their estimates of global zoonotic disease emergence. Whilst these improvements are much welcomed and represent an important step forward for this work I do have some concerns about their appropriateness and relevance to some of the outputs. There are a large number of issues that I would like to see considered - I would advocate for major revisions over rejection however as I feel this methodology represents an important advance over what is already considered a highly influential paper. Perhaps some of these issues are confusion on my part - if so there may need to be some more of the methods outlined earlier in the paper (an issue common to many articles where the methods are placed at the end). In particular, I am unclear how the Figure outputs and bias layer interact (which may negate some of the comments below).

Major comments:

- The authors should explicitly cite what definition of "emerging infectious disease" they are using at the outset, as very many different pathogens are classed as EIDs for a variety of reasons. The dataset clearly references the origin event for the pathogen and hence inherently uses the EID definition referring to novelty. This is a particularly important aspect to state early on in the introduction (and it is discussed in more detail in the attached supplementary paper submitted elsewhere). As a result of this distinction, I think it may perhaps be wise to not mention the West African Ebola outbreak (lines 37) as an example of a "newly emerged pathogen" therefore,

particularly as the Ebola information within the database used for this analysis only has the 1976 index case from Yambuku. This may therefore cause additional confusion. As I said, a clear statement of what emerging infectious disease means, and what components of that definition are relevant here, would clarify any potential issues.

- I am unsure how the temporal component of the covariate layers was included. If the covariate value extracted for each occurrence was linked to the time at which the pathogen emerged, more information needs to be included on how the inter-decadal differences were assigned to mid-decade years, or how a specific year was assigned to a GRUMP 5-year bin? If one synoptic value for each covariate was used (i.e. two occurrences in the same location, but separated by thirty years, are assigned the same covariate values), this needs to be addressed as this is likely simplifying large differences in the data - for many covariates the full time period is measured, so why can it not be used? I believe a hybrid of these approaches was used (wherever possible an occurrence year was matched to an annual covariate value) but this needs to be explicitly mentioned and specific assignment of values discussed for many of the aggregated variables.

- A lot more information needs to be provided in the Observation bias dataset generation as this in its own right represents a major output. I have several concerns as to whether the layer is relevant for describing this issue as well as how it was generated and then subsequently used. It is unclear how the Python script parses geographic information without looking at the code; I assume that only the main text was used rather than the author addresses etc.? Furthermore, how are incidental geographies dealt with (i.e. locations mentioned in passing in the introduction or discussion) versus actual geographic information relevant to the bias layer (i.e. the location where the disease was reported)? How are oddities where geographic names and common terms overlap dealt with e.g. Marburg used for Marburg virus disease .v. Marburg (the settlement)? How are geographic events not related to diseases dealt with (e.g. the Rio Olympics and potential disease risk? - are these all marked as Rio de Janeiro?). What languages were and weren't able to be scraped; what different biases does this introduce if Arabic, or Chinese language papers cannot be easily extracted? An independent cross validation of the accuracy of this mass extraction should be presented where the scraper outputs are compared to the geographies extracted by a secondary method (i.e. human extraction). I can see many pitfalls in this approach - hopefully the authors have dealt with these issues and omitted them from the manuscript due to space constraints. This would be great to see as an SI.

- The methods and results for the bias layer boosted regression tree should be presented in far more detail as this represents a key modelled output of this study. Whilst I have a good understanding of how the BRT for the emergence events behaves, apart from an R-squared value, I have no information on the relative contribution of the covariates etc. What were the parameters used on this BRT (what was the number of trees used etc.). More detailed information should be provided and should be represented in a separate figure

- There will be uncertainty associated with this bias metric - how was this subsequently propagated into the bias correction component of the study?

- I am surprised at the covariates used to gauge the observation bias as these strike me as only tangentially related? Why is there no discussion as to how distance to hospital, diagnostic capacity etc. could be included or were omitted? Even something such as an urban accessibility layer, which exists at a high spatial resolution, or wealth indices such as G-Econ from Yale could be considered

- I would say these two are likely far better correlates of observation bias than certain land cover types.

- I am very concerned that there is a significant bias introduced into the EID model by using the same covariates in the observation bias model. Given that polygonal data as well as background data was weighted based upon this observation bias layer, surely there is double dipping of covariate information for that subset? There may be unintentional skewing of the presence dataset based upon the values of these layers -there will therefore be a subsequent bias in the covariate relationship determined in the EID model. I am not convinced that this is an insignificant impact - this could be clarified by (a) presenting more information on the observation bias layer and (b) demonstrating how a different sampling procedure for both polygons and background data (such as random) causes differences (or not) in the subsequent outputs. The current system suggests to me potential circularity, and therefore it is unsurprising that many of the bias covariate layers

result as main predictors in the final model.

- I think seeing more information on the bias model will help clarify this, but it is odd to note that, once bias is factored out, many capital cities remain at high risk of zoonotic disease emergence, something that isn't addressed in the current narrative of this paper. London, Madrid, Paris, Moscow, Baghdad, Tehran and a large number of other global major cities come up (or remain) as high risk. This isn't really addressed in the discussion, which is focused more on mammalian biodiversity and perturbation. Why would Berlin or Helsinki (which go down in risk to effectively zero from Figure 3A to 3B) differ significantly from Moscow in this respect? As I said, seeing how the bias layer behaves would help inform this.

- What were the tuning parameters used in both BRT models? What were number of trees, tree complexity, learning rate, bag fraction, step size etc. This is assuming some variant of the basic dismo implementation of BRT is used? Other setups will have different variations but listing their values is still important.

- I think the use of "risk" requires clarification - the output of the BRT is a measure of environmental similarity between a pixel of interest and a hypothesized "ideal" emerging infectious disease event environmental profile [here environmental is being used as short hand for all the covariates used, some of which are in this study biotic rather than abiotic in nature]. A true measure of "risk" would therefore require a quantification of all the other factors that are not considered in the analysis. Their omission is understandable given the lack of information, but the authors must therefore explicitly state somewhere that this environmental similarity index correlates or approximates to risk. The sentence in line 406 implies the terms are synonymous, which is not the case.

- The authors utilize the partial dependency plots to make some strong assertions concerning relationships. These plots act as good guides for directions of trends (i.e. positive or negative associations) but are typically weak when used in a quantitative manner (as per the statements in lines 83/84). I quote Elith et al. 2008 "A working guide to boosted regression trees" "Visualization of fitted functions in a BRT model is easily achieved using partial dependence functions that show the effect of a variable on the response after accounting for the average effects of all other variables in the model. While these graphs are not a perfect representation of the effects of each variable, particularly if there are strong interactions in the data or predictors are strongly correlated, they provide a useful basis for interpretation." The authors have not considered the potential for strong correlations within the dataset that could influence this result. As a consequence, I believe that the strong wording on association of specific changes in mammalian species richness and precise values be downplayed or omitted. The authors have however presented a large amount of information on the partial dependence plots (both uncertainty and input data spread) which was great to see. I still feel however that it could be misinterpreted when certain relationship claims are stated (over others that are equally plausible given the modelled output) and not immediately caveated with the uncertainty complications touched upon, for instance, in line 164.

- The authors claim (line 127/8) that their analysis supports the general EID narrative of land use change driving zoonotic disease emergence - this seems to me only tenuously supported by the pasture change dataset - other change layers (such as population change and cropland) are lower and therefore I don't believe this statement is supported. The statement in line 236-238 that the map "shows the global distribution of zoonotic EID risk is concentrated in tropical regions where wildlife biodiversity is high, human population dense and growing and land use change occurring rapidly" does not conform with the outputs of the BRT - indeed the model is relatively unaffected by population size (small populations in fact have a much higher response), population change (where increasing population has led to a drop in relative probability) and land use change has a mixed impact (loss of pasture has high relative probability, but only substantial cropland change has an impact). Admittedly, I presume many of these drop out after correction for the bias layer (I'm unclear whether Figures 1 and 2 are before or after the bias correction step), but I cannot judge this without seeing how these covariates are used within the bias layer.

Minor comments

- The direction of the weighting for random selection of background points needs to be explicitly

stated - did high observation areas receive more, or less, background points? The implications of this are important (high observation areas receiving more background points would conform to the more widely accepted generation approach)

- Can the authors provide more information on the equation outlined in lines 442/443? What is "per person risk of event"? Is it the output of the BRT divided by the population; if so I don't believe that output accurately reflects that statement without a major caveat (see major comment above)

- How is relative probability in Figure 2 calculated? There is an important translation step from the start output of such BRTs (marginal effects) to this relative probability measure - is this what is outlined in lines 469-472?

- What was the protocol for inclusion of new EID occurrences? A brief synopsis of the associated submitted paper describing the EIDR would be welcomed

- Why was 1970 used as the cutoff date? What impact on the results does changing this cut-off date cause? The date being tied to one of the covariate layers makes sense from a pragmatic point of view, but it is still important to gauge what impact this might have.

- In Figure 3, please color Greenland appropriately (not sure why it is grey) and divide Sudan into its current two geographies (and check that an up-to-date national boundary layer is being used). Similarly, Serbia looks to be colored the same as water, not a (presumed) green.

Reviewer #3 (Remarks to the Author):

This paper updates a global database of emerging infection disease events and develops a spatial model of these events in relation to potential demographic, environmental and biological determinants. The paper confirms prior findings that EID risk is elevated in tropical areas, where wildlife biodiversity is high and human populations are growing. A major claim is that the authors have developed a novel way to correct for geographical biases in EID event data and that this results in improved global hotspot maps for EIDs. The authors suggest that this map could be used to target surveillance and pathogen discovery.

The key novelty of the paper is in the novel correction for observation bias and in the examination of non-linear relationships between EID events and potential determinants like biodiversity. The latter enhances the biological interpretability of the ecological and environmental effects, for example offering potential to distinguish the mechanisms by which biodiversity enhances or reduces transmission. The updating of the database and refinement of the other spatial predictors are important but quite incremental advances upon the Jones et al. 2008 paper.

The paper will influence methodologies to account for sampling bias and appropriate ecological and social determinants for mapping and understanding of global disease distributions. It also encourages more careful interpretation of the impacts of biodiversity and land use and population changes on disease patterns.

The conclusions could be strengthened considerably with refinements to the BRT methodology and more careful interpretation of some of the effects. In relation to the BRT methodology, I would suggest that the ratio of pseudo-absences to presences is too high according to best practice from simulation studies. The paper below suggests that when using BRT it is optimal to use an equal number of pseudo-absences to presences.

Barbet-Massin, M., Jiguet, F., Albert, C.H., and Thuiller, W. (2012). Selecting pseudo-absences for species distribution models: how, where and how many? *Methods in Ecology and Evolution* 3, 327-338.

Here, the authors have used 100 times as many pseudo-absences as presences (page 29) which will inflate their AUC metrics which are quite low for models of this kind. It would be better for the authors to also present TSS to evaluate their model which is independent of prevalence (see Barbet-Massin paper) and to use an equal ratio of absences to presences.

I would like to see some assessment of collinearity between predictor variables. Although BRTs are quite robust to collinearity, you may still find that variable contributions are being spread across

collinear sets, making it difficult to interpret the final model and separate individual variable effects. For example pasture coverage is likely to be highly correlated with large livestock densities. I would also expect the mammal diversity layers to be highly correlated with the climate-based global environmental stratification. It might be that mammal diversity is being selected because it better represents important broad-scale climate variability than a categorical stratification. The authors could test this by offering alternative continuous climate gradients to the analysis. Were interactions between variables, like between population density and population change or between livestock density and mammal diversity considered explicitly in the BRT modelling?

I highly appreciate the efforts to correct for observation bias. However, the authors should give a more detailed treatment of the meaning and potential shortcomings of their observation bias measure. Is the open-access subset biased towards researchers in richer locations that can pay open-access fees? Or is the subset spatially biased towards more recent research effort, mismatching with the temporal range of the EID data? The authors should specify the temporal range of the publication effort data on page 22. Also, are the processes that lead to an EID event making it into the EID database, similar to the process that leads to an EID event being written up for publication. I assume most disease events are not initially notified by public health researchers but by public health authorities so are the suites of sites the same? To maximise the impact of this methodological improvement on the field, I would like to see the authors deal briefly with some of these issues and how they may impact their conclusions. Overall, I agree it is probably an advance over the prior metrics for observation bias referenced on page 22 and the impacts on the heatmap in Fig. 3 are interesting and important. It may be interesting to include in the supplementary methods part some brief mention of the key environmental drivers of publication effort, according to the BRT model that they used to smooth the raw data (pgs. 22-23).

Some claims are overstated on the basis of this particular analysis. I am not convinced from the explanation below that the authors can really distinguish between the "perturbation" of pathogen dynamics versus the exposure to a novel pathogen pool from this analysis. Isn't the amount of edge interface between natural and human modified habitats rather than the overall cover change the relevant landscape metric to disentangle these two?

"Inter-decadal pasture, cropland and human population change show less relative influence in explaining global patterns of EID occurrence. Their partial dependence plots, however, suggest increased EID event probability at both positive and negative values. This supports the hypothesis that land use change may cause "perturbation" of pathogen dynamics in a system, promoting cross-species transmission, as opposed to representing exposure to a novel pathogen pool due to encroachment on previously undisturbed habitat 10. Perturbation may increase emergence risk by impacting the number of infectious hosts, or altering pathogen prevalence as a function of habitat intactness by a range of mechanisms related to meta-community ecology 26, whereas we would not expect risk to also increase with decreasing pasture change under a novel pathogen pool explanation."

Apart from the absence sampling, accuracy statistics and collinearity problems discussed above, the statistics are appropriate, including in the treatment and presentation of the uncertainty in the analysis. There is insufficient detail about the BRT for replication of the study. The authors should add the learning rate, tree complexity and bag rate settings of the BRT in the supplementary methods section.

Specific comments

Page 3. Please reiterate the dependent variable for readers unfamiliar with 4.

Page 11. Move line about " increase in zoonotic EID relative risk at low levels of mammal species richness" to the results.

Page 12. Some elements of this discussion are quite speculative and not particularly well linked to the results of this particular analysis.

For example, this statement seems rather obvious: "Indeed, both positive and negative effects of wildlife richness on disease risk could be occurring, depending, for example, on geographic region, spatial scale, subgroups of diseases, host species composition, or a host of other factors."

Page 12. "increase with decreasing pasture change under a novel pathogen pool explanation.". Shouldn't "decreasing pasture change"

The sections on model impacts and future work to be based on this framework are overlong and could be substantially reduced.

Reviewers' comments:

Reviewer #1 (Remarks to the Author):

This is a straightforward report of a new interactive data base on emerging infectious diseases. As such it is not a typical example of primary research, but the database is of potential importance to researchers and students.

There is little to criticize in this straightforward and brief description. I found it most helpful to go to the website and "play" with various parameters as I was reading the ms.

The title is a bit misleading as it does not indicate that the ms. describes website data and how to access and use it. Also, I must say, the actual writing is dry and not very interesting, whereas the website itself is quite interesting. Perhaps there is a way to put more pizzazz into the text.

In our first submission, we included a manuscript with methods, results and discussion detailing analyses of correlates of emerging infectious disease events (EID events). We have previously collated these EID events into a database which is the subject of a descriptive paper submitted elsewhere. As per Instructions to Authors for *Nature Communications*, we included a copy of that paper. Based on the (fair and appropriate) comments from reviewer #1, it seems that the reviewer may have read our database appendix as the main document. We apologize for any confusion, and in our re-submission, we have made an effort to be far more explicit about the differences between the paper describing an updated version of our database (EIDR), and the analyses we have conducted here. We have also made sure that the writing will appeal to readers of broad interests as suggested by this reviewer.

The term "emergence" in their text is used to include "re-emergence" as well, and this should perhaps be explicitly spelled out.

We have now clearly laid out the rationale for excluding re-emergence events, and defined our terminology more clearly throughout, as suggested by this and other reviewers.

One difficulty in assessing the power of the site data is that if anything has been missed how would one know it?

Attempts were made to be very thorough in the literature review and we have laid out (in the methods) in more detail how this was done. We have also developed a novel reporting effort metric that we use to incorporate the effect of unequal detection probabilities of EID events due to lack of research effort.

Also there must be some arbitrariness to the criteria for including a disease as newly emerging, and this is not very clear. For example, is every new salmonella type, or every new strain of an evolving virus a

new disease. Was, for example, Bundibugyo ebolavirus disease an EID or not, and how is that decided? What about drug resistant viruses? Are they treated differently than drug resistant bacteria?

We have re-written in detail our definitions and criteria for including diseases as emerging. We only consider zoonoses originating in wildlife in this paper, therefore drug-resistant viruses are likely not important in the current analysis.

It is curious that they did not try to include past emergences, from before 1940. Hopefully that can be added at a later date.

The reasons we begin our database in 1940 were originally laid out in Jones *et al.* (2008), based on the Institute of Medicine's (1997 and 2001 reports) examples of currently or recently emerging diseases which likely originated within this timescale. We do envisage future efforts to trace EIDs back further than 1940, but due to paucity and reliability of data, this effort will likely be a long-term project beyond the current scope of work.

Finally, I think readers will be skeptical about possible subjectivity in assigning disease drivers such as ecosystem change or war.

We agree with the reviewer that assignment of drivers likely has a great deal of subjectivity. However, we have tried to maintain the 'gold standard' of the deliberations of the IOM's Forum on Microbial Threat's reports, which assign these drivers to different EIDs, at the least in order to test their veracity. Future work will clarify these, but again is likely beyond the scope of the current paper.

Reviewer #2 (Remarks to the Author):

In this article the authors outline a series of major advances to the methodology of Jones et al. 2008 and update their estimates of global zoonotic disease emergence. Whilst these improvements are much welcomed and represent an important step forward for this work I do have some concerns about their appropriateness and relevance to some of the outputs. There are a large number of issues that I would like to see considered - I would advocate for major revisions over rejection however as I feel this methodology represents an important advance over what is already considered a highly influential paper. Perhaps some of these issues are confusion on my part - if so there may need to be some more of the methods outlined earlier in the paper (an issue common to many articles where the methods are placed at the end). In particular, I am unclear how the Figure outputs and bias layer interact (which may negate some of the comments below).

We thank the reviewer for their comments. We have revised and improved many parts of our methodology, and have attempted to clarify them in the body of the paper (and additional Supplementary Information provided). We hope that our explanations of the methods are clearer. Although we are generally constrained by the methods coming *after* the results, we have, at the risk

of being duplicative, clarified concepts on first use where possible (such as the following point on the definition of “EID events”) earlier in the manuscript.

Major comments:

- The authors should explicitly cite what definition of "emerging infectious disease" they are using at the outset, as very many different pathogens are classed as EIDs for a variety of reasons. The dataset clearly references the origin event for the pathogen and hence inherently uses the EID definition referring to novelty. This is a particularly important aspect to state early on in the introduction (and it is discussed in more detail in the attached supplementary paper submitted elsewhere). As a result of this distinction, I think it may perhaps be wise to not mention the West African Ebola outbreak (lines 37) as an example of a "newly emerged pathogen" therefore, particularly as the Ebola information within the database used for this analysis only has the 1976 index case from Yambuku. This may therefore cause additional confusion. As I said, a clear statement of what emerging infectious disease means, and what components of that definition are relevant here, would clarify any potential issues.

We have included an expanded definition of EID events in the Methods, and refer to our working definition of an ‘event’ earlier in the paper, in the introduction. We have also changed the West African Ebola example, per the reviewer’s suggestion.

- I am unsure how the temporal component of the covariate layers was included. If the covariate value extracted for each occurrence was linked to the time at which the pathogen emerged, more information needs to be included on how the inter-decadal differences were assigned to mid-decade years, or how a specific year was assigned to a GRUMP 5-year bin? If one synoptic value for each covariate was used (i.e. two occurrences in the same location, but separated by thirty years, are assigned the same covariate values), this needs to be addressed as this is likely simplifying large differences in the data - for many covariates the full time period is measured, so why can it not be used? I believe a hybrid of these approaches was used (wherever possible an occurrence year was matched to an annual covariate value) but this needs to be explicitly mentioned and specific assignment of values discussed for many of the aggregated variables.

We thank the reviewer for pointing out the parts of this methodology which lacked clarity. In the case of the specific concerns:

- **Two occurrences in the same location but at different years would each receive the appropriate temporal covariate, as would associated background points.**
- **To assign values, years were rounded to the nearest decade (mid-decade years were rounded up); for the “change” layers that represented differences between different decades, events were assigned to the layer for their decade (e.g. 1994 would be matched with the population layer from 1990 and the population change layer for 2000 – 1995).**
- **The reviewer is correct that we used a hybrid approach, for valid reasons – mainly that many datasets are only available as static layers.**

We have added a column to Table 1 to indicate which variables were included as static vs. temporal. In addition, we have reworked the section of our Methods describing our resampling regimes, to reflect updated methods from all reviewers. We hope this new section is clearer.

- A lot more information needs to be provided in the Observation bias dataset generation as this in its own right represents a major output. I have several concerns as to whether the layer is relevant for describing this issue as well as how it was generated and then subsequently used. It is unclear how the Python script parses geographic information without looking at the code; I assume that only the main text was used rather than the author addresses etc.? Furthermore, how are incidental geographies dealt with (i.e. locations mentioned in passing in the introduction or discussion) versus actual geographic information relevant to the bias layer (i.e. the location where the disease was reported)? How are oddities where geographic names and common terms overlap dealt with e.g. Marburg used for Marburg virus disease .v. Marburg (the settlement)? How are geographic events not related to diseases dealt with (e.g. the Rio Olympics and potential disease risk? - are these all marked as Rio de Janeiro?). What languages were and weren't able to be scraped; what different biases does this introduce if Arabic, or Chinese language papers cannot be easily extracted? An independent cross validation of the accuracy of this mass extraction should be presented where the scraper outputs are compared to the geographies extracted by a secondary method (i.e. human extraction). I can see many pitfalls in this approach - hopefully the authors have dealt with these issues and omitted them from the manuscript due to space constraints. This would be great to see as an SI.

We appreciate the reviewer's insightful and valid comments on this issue. The reviewer is correct that our reporting effort dataset is now more complex (and we believe more defensible and rigorous) than that used in Jones *et al.* (2008) and similar studies (e.g. Yang *et al.* (2012)). We now have more fully documented the reporting effort dataset methods in the Supplementary Methods: PubCrawler. The new Supplementary Information section includes a more detailed explanation of the PubCrawler natural language processing (NLP) algorithm, the Boosted Regression Trees model used, a discussion of the limitations of the method, and figures describing the model and comparing it to previous methods.

In the revision, we also updated the reporting effort layer used in the model, reflecting updates to the PubCrawler NLP algorithms. The new layer is very similar to the old layer in its distribution, despite revisions we made to the code (R^2 between original and revised reporting effort outputs = 0.85).

We plan to continue to refine the PubCrawler NLP code, and we are working on a separate publication detailing an extended and generalized version. The code itself is open-source under an Apache 2.0 license: (<https://github.com/ecohealthalliance/pubcrawler> and <https://github.com/ecohealthalliance/annie> are the two main repositories needed to replicate the publication layer).

Addressing the specific items raised, the reviewer is correct that the main text of published articles was used rather than authors addresses to identify geographic location (clarified in the revision). This

corrects one of the likely erroneous assumptions of Jones *et al.* (2008) that an author's address correlates with the place of study. We have fully addressed this and the reviewer's other specific concerns, e.g. incidental vs. actual geographies, geographic terms in common names, and languages in data scraped, in the new Supplementary Methods: PubCrawler.

Briefly, we do not have an adequate gold standard dataset for this layer at this point in time, so are unable to measure its accuracy using standard statistical validation tests. However, a Poisson GLM regressing the new publication index (aggregated to the county level) against the count of publications in the Jones *et al.* layer indicates a statistically significant association between the two layers, with a McFadden's Pseudo R^2 of 0.78. Since the new measure represents a new process that is theoretically aligned with our outcome, and produces a similar distribution at country level to the old index, we think it is a meaningful advancement in disaggregating infectious disease reporting effort past country level. We have added a discussion of this to the SI.

- The methods and results for the bias layer boosted regression tree should be presented in far more detail as this represents a key modelled output of this study. Whilst I have a good understanding of how the BRT for the emergence events behaves, apart from an R-squared value, I have no information on the relative contribution of the covariates etc. What were the parameters used on this BRT (what was the number of trees used etc.). More detailed information should be provided and should be represented in a separate figure

We have included detailed information on the BRT parameters used in all models discussed, and included full details of the reporting effort model in Supplementary Methods: PubCrawler.

- There will be uncertainty associated with this bias metric - how was this subsequently propagated into the bias correction component of the study?

We acknowledge that there will be uncertainty with the reporting effort estimate, adding to the numerous other sources of uncertainty that we have made extensive efforts to tackle robustly and transparently in this study. We include and present our reporting effort procedure here primarily as a conceptual and methodological advance over the original bias layer used in Jones *et al.*, Yang *et al.*, and other studies that face this difficult issue but have been forced to use country-level data. To that end, we have included a discussion in our ms of the relative advance this represents, as well as its limitations. Longer term, we recognize the utility of our approach for numerous applications and have a strategy in place to continue to develop our methods and plan to publish a stand-alone reporting effort package that is generalizable and validated with gold-standard (human-extracted) data.

- I am surprised at the covariates used to gauge the observation bias as these strike me as only tangentially related? Why is there no discussion as to how distance to hospital, diagnostic capacity etc. could be included or were omitted? Even something such as an urban accessibility layer, which exists at a high spatial resolution, or wealth indices such as G-Econ from Yale could be considered - I would say

these two are likely far better correlates of observation bias than certain land cover types.

In the new version of our reporting effort layer, we have modified the predictor set used. We included measures of population, urban extent, accessibility / remoteness, health burden (DALYs), healthcare expenditure, and GDP. This model is now detailed more comprehensively in the Supp. Info section about the reporting effort layer. The resolution of this layer matches the resolution of our study grid (~100km), which although a big improvement over country level data, is still not high enough to warrant inclusion of some of the finer scale predictors (e.g., urban accessibility) suggested by the reviewer. However, we are confident that the predictor set chosen will usefully proxy some of these factors at our study's spatial scale. We also compared the outputs of the original and revised reporting effort layers, and these were highly correlated ($R^2 = 0.85$).

- I am very concerned that there is a significant bias introduced into the EID model by using the same covariates in the observation bias model. Given that polygonal data as well as background data was weighted based upon this observation bias layer, surely there is double dipping of covariate information for that subset? There may be unintentional skewing of the presence dataset based upon the values of these layers -there will therefore be a subsequent bias in the covariate relationship determined in the EID model. I am not convinced that this is an insignificant impact - this could be clarified by (a) presenting more information on the observation bias layer and (b) demonstrating how a different sampling procedure for both polygons and background data (such as random) causes differences (or not) in the subsequent outputs. The current system suggests to me potential circularity, and therefore it is unsurprising that many of the bias covariate layers result as main predictors in the final model.

We improved and simplified our admittedly complex set of sampling regimes, and clarified our explanations, based on the comments of Reviewers 2 and 3. For our bootstrap resampling regime, we resampled events with replacement uniformly, and the reporting effort layer was then used to do the sampling of a grid cell *within events*. This means that the overall distribution of events contributes to the overall shape of the distribution; indeed, if an event was only present in one grid cell, that grid cell would be the only grid cell sampled. In the case of events with less certainty, the reporting effort layer essentially serves as a prior.

One way of thinking about this is to imagine two contrasting EID events, one was known to have occurred in a single grid cell, and another which was known to have occurred somewhere on the surface of the planet. If we weight our background sample by reporting effort but fail to weight within each EID event by the same variable, our loosely-bounded EID event will more often fall in areas of low reporting effort and *induce* bias into the dataset. Weighting the presence sample makes no difference in the single-grid-cell case.

Though this is effectively the same as our prior way of weighting samples, it is clearer, as we now address sampling error (via bootstrap) and location uncertainty (sample within event polygon, weighted by the prior of reporting effort) at separate stages of sampling. We have included figures in the supplementary information (Supplementary Notes – Supplementary Figures 3.1 and 3.2) to

illustrate this process and its effects on our outcome of interest (EID events), showing the aggregated weight of sampling probability for each grid cell with and without the weighting (plus a version of the plots with log-transformed color palette scaling, to reveal low-weighted grid cells).

- I think seeing more information on the bias model will help clarify this, but it is odd to note that, once bias is factored out, many capital cities remain at high risk of zoonotic disease emergence, something that isn't addressed in the current narrative of this paper. London, Madrid, Paris, Moscow, Baghdad, Tehran and a large number of other global major cities come up (or remain) as high risk. This isn't really addressed in the discussion, which is focused more on mammalian biodiversity and perturbation. Why would Berlin or Helsinki (which go down in risk to effectively zero from Figure 3A to 3B) differ significantly from Moscow in this respect? As I said, seeing how the bias layer behaves would help inform this.

We have expanded our explanation to why we think this might be happening (the section on factoring bias out. We explicitly do not just divide by reporting effort; we multiply by population density.

One line of reasoning is that reporting *effort* both includes the true multiplicative effect of population density on EID event risk (i.e. if there are twice as many people in a location, there would be twice as many disease events, modulo any other thresholding effect) and “bias”, where the distribution of reporting effort differs from the distribution of population density. An assumption is made that a unit of reporting effort translates to the probability of detecting a disease event in a person. In this case, the assumed “optimal” distribution of reporting effort to detect EID events — disease events in the human population — is that of the human population distribution.

This goes to say, when we factor our reporting *effort* layer out, we must factor this “ideal reporting effort” layer back in, (and any baseline multiplicative effect against which the model was fit) by multiplying by human population. This is probably why cities appear high risk here. And, to visual inspection, the raw data (Supp. Notes – Supp. Figs 3.1 and 3.2) do show that observations of EID events are higher in areas with more people, and we do not think this is entirely due to bias.

The partial dependence plots for the weighted model show a somewhat declining effect of population on the EID risk index — relative to reporting effort (i.e. population density and bias).

Upon review, we realized that we had used the terms “reporting effort” and “observation bias” interchangeably, whereas the former more accurately describes the measure we created (the latter referring to the difference between the actual and optimal distribution of reporting effort). We have changed our language to be more consistent and clear on this point throughout the MS and Supplementary Information.

We have also added another figure to the paper, showing a map of the model’s output after factoring reporting effort out, but before reweighting by human population density. We feel that this helps illustrate our point, and — to the reviewer’s point — cities are not points of high effect in this map.

- What were the tuning parameters used in both BRT models? What were number of trees, tree complexity, learning rate, bag fraction, step size etc. This is assuming some variant of the basic dismo implementation of BRT is used? Other setups will have different variations but listing their values is still important.

We used an extension of the R package dismo to simplify our resampling regime. We have now provided thorough information on all the tuning parameters used in the main manuscript.

- I think the use of "risk" requires clarification - the output of the BRT is a measure of environmental similarity between a pixel of interest and a hypothesized "ideal" emerging infectious disease event environmental profile [here environmental is being used as short hand for all the covariates used, some of which are in this study biotic rather than abiotic in nature]. A true measure of "risk" would therefore require a quantification of all the other factors that are not considered in the analysis. Their omission is understandable given the lack of information, but the authors must therefore explicitly state somewhere that this environmental similarity index correlates or approximates to risk. The sentence in line 406 implies the terms are synonymous, which is not the case.

We thank the reviewer for pointing this out. We tried to adopt a *lingua franca* somewhere between the terminologies of epidemiology and ecology, and in the current version we have made another attempt to clarify our use of the term 'risk'. We now refer to our model's output as a "risk index", to indicate that it is not an absolute measure of risk / probability but a relative measure, correlated with risk. We hope this is more succinct and clearer. Where we do use "risk", we preface it with "EID risk index, henceforth 'EID risk'", and even then only use "risk" by itself when talking about it being elevated in areas i.e. in a relative manner.

- The authors utilize the partial dependency plots to make some strong assertions concerning relationships. These plots act as good guides for directions of trends (i.e. positive or negative associations) but are typically weak when used in a quantitative manner (as per the statements in lines 83/84). I quote Elith et al. 2008 "A working guide to boosted regression trees" "Visualization of fitted functions in a BRT model is easily achieved using partial dependence functions that show the effect of a variable on the response after accounting for the average effects of all other variables in the model. While these graphs are not a perfect representation of the effects of each variable, particularly if there are strong interactions in the data or predictors are strongly correlated, they provide a useful basis for interpretation." The authors have not considered the potential for strong correlations within the dataset that could influence this result. As a consequence, I believe that the strong wording on association of specific changes in mammalian species richness and precise values be downplayed or omitted. The authors have however presented a large amount of information on the partial dependence plots (both uncertainty and input data spread) which was great to see. I still feel however that it could be misinterpreted when certain relationship claims are stated (over others that are equally plausible given the modelled output) and not immediately caveated with the uncertainty complications touched upon, for instance, in line 164.

This is an excellent and useful comment - we have toned down our interpretations based on the partial dependence plots, sticking to commenting on the overall shape of the effects, the uncertainty bounds, and the order of importance of the variables.

- The authors claim (line 127/8) that their analysis supports the general EID narrative of land use change driving zoonotic disease emergence - this seems to me only tenuously supported by the pasture change dataset - other change layers (such as population change and cropland) are lower and therefore I don't believe this statement is supported. The statement in line 236-238 that the map "shows the global distribution of zoonotic EID risk is concentrated in tropical regions where wildlife biodiversity is high, human population dense and growing and land use change occurring rapidly" does not conform with the outputs of the BRT - indeed the model is relatively unaffected by population size (small populations in fact have a much higher response), population change (where increasing population has led to a drop in relative probability) and land use change has a mixed impact (loss of pasture has high relative probability, but only substantial cropland change has an impact).

Admittedly, I presume many of these drop out after correction for the bias layer (I'm unclear whether Figures 1 and 2 are before or after the bias correction step), but I cannot judge this without seeing how these covariates are used within the bias layer.

This is a very good point, and we have made an effort to clarify our interpretation in the revision. Figures 1 and 2 are the effect of the model relative to the reporting effort layer. We have included in SI similar figures for the supplementary model run without the reporting effort layer, and tried to make our language around this clearer.

Minor comments

- The direction of the weighting for random selection of background points needs to be explicitly stated - did high observation areas receive more, or less, background points? The implications of this are important (high observation areas receiving more background points would conform to the more widely accepted generation approach)

The reviewer is correct; areas with a higher reporting effort score are up-weighted. We have clarified this and the other points suggested below in the main manuscript.

- Can the authors provide more information on the equation outlined in lines 442/443? What is "per person risk of event"? Is it the output of the BRT divided by the population; if so I don't believe that output accurately reflects that statement without a major caveat (see major comment above)

We did not intend to imply that we were able to estimate the absolute per-person risk of EID event. In the descriptive equations, we were careful to use the \propto symbol to imply "directly proportional to", but have changed "risk" to say "relative risk" to be more explicit. We have simplified and reworked the equations to more clearly state our point (the same point as is outlined in a response to a prior

question). We have also revised our language to refer to the outcome as a risk index, as described above.

- How is relative probability in Figure 2 calculated? There is an important translation step from the start output of such BRTs (marginal effects) to this relative probability measure - is this what is outlined in lines 469-472?

We present the response, not the direct link function. In addition, we have adjusted our plots slightly, both in response to other reviewer suggestions about our sampling regimes and our observation that the ability to perceive the relevant part of the response curves is thrown off by the x axes being scaled to the minimum and maximum observed values, which only occur in a few of the sampling runs. We have truncated the x axes, per variable, by (0.1, 0.9) quantiles, and presented the full-range plots in SI. Due to differences in sampling regime (in response to Reviewer 3) we have also rescaled our Y axes so that the middle of the plot is fixed at the line of no effect.

- What was the protocol for inclusion of new EID occurrences? A brief synopsis of the associated submitted paper describing the EIDR would be welcomed

We have included details of this in the methods.

- Why was 1970 used as the cutoff date? What impact on the results does changing this cut-off date cause? The date being tied to one of the covariate layers makes sense from a pragmatic point of view, but it is still important to gauge what impact this might have.

This cutoff was used to explicitly address a shortcoming of the original Jones *et al.* (2008) paper. We used 1970 as a cutoff because 1) where we had different temporal layers for important variables, 1970 was the earliest; we would not have been able to match earlier events with relevant temporal data were we to include older events. 2) from an empirical observation of the original data, those events from 1970 to 2008 represent about 2/3 of the data, and gave us reasonable power for our analyses.

We have included in Supplementary Notes a plot of the distribution of excluded events. We are unable to assess how the geographic distribution of drivers have changed prior to 1970 due to the paucity of data available before that time.

- In Figure 3, please color Greenland appropriately (not sure why it is grey) and divide Sudan into its current two geographies (and check that an up-to-date national boundary layer is being used). Similarly, Serbia looks to be colored the same as water, not a (presumed) green.

We have corrected these. Greenland and Serbia are no longer underwater. Sudan uses correct outlines.

Reviewer #3 (Remarks to the Author):

This paper updates a global database of emerging infection disease events and develops a spatial model of these events in relation to potential demographic, environmental and biological determinants. The paper confirms prior findings that EID risk is elevated in tropical areas, where wildlife biodiversity is high and human populations are growing. A major claim is that the authors have developed a novel way to correct for geographical biases in EID event data and that this results in improved global hotspot maps for EIDs. The authors suggest that this map could be used to target surveillance and pathogen discovery. The key novelty of the paper is in the novel correction for observation bias and in the examination of non-linear relationships between EID events and potential determinants like biodiversity. The latter enhances the biological interpretability of the ecological and environmental effects, for example offering potential to distinguish the mechanisms by which biodiversity enhances or reduces transmission. The updating of the database and refinement of the other spatial predictors are important but quite incremental advances upon the Jones et al. 2008 paper.

The paper will influence methodologies to account for sampling bias and appropriate ecological and social determinants for mapping and understanding of global disease distributions. It also encourages more careful interpretation of the impacts of biodiversity and land use and population changes on disease patterns.

We thank the reviewer for these positive comments, and have attempted to include much more detail on our measure of reporting effort in the hopes that we can provide value to future research projects in this area.

The conclusions could be strengthened considerably with refinements to the BRT methodology and more careful interpretation of some of the effects. In relation to the BRT methodology, I would suggest that the ratio of pseudo-absences to presences is too high according to best practice from simulation studies. The paper below suggests that when using BRT it is optimal to use an equal number of pseudo-absences to presences.

Barbet-Massin, M., Jiguet, F., Albert, C.H., and Thuiller, W. (2012). Selecting pseudo-absences for species distribution models: how, where and how many? *Methods in Ecology and Evolution* 3, 327-338.

Here, the authors have used 100 times as many pseudo-absences as presences (page 29) which will inflate their AUC metrics which are quite low for models of this kind. It would be better for the authors to also present TSS to evaluate their model which is independent of prevalence (see Barbet-Massin paper) and to use an equal ratio of absences to presences.

We thank the reviewer for these useful comments and suggestions. We have updated our sampling regimes (and simplified them slightly) after reviewing this paper and others.

- **We now conduct bootstrap resampling, drawing a single presence and absence point for each event, running 1000 iterations.**

- **We have replaced our 4-fold partitioning and cross-validation with exhaustive leave-one-out cross-validation, running 10 iterations per event, and calculating the True Skill Score (TSS), as suggested. We report this and the AUC. For our weighted model, we report both relative to the null used to weight the model and to a uniform (across the Earth's surface) null. For our unweighted model we present both statistics relative to a uniform null.**

I would like to see some assessment of collinearity between predictor variables. Although BRTs are quite robust to collinearity, you may still find that variable contributions are being spread across collinear sets, making it difficult to interpret the final model and separate individual variable effects. For example pasture coverage is likely to be highly correlated with large livestock densities. I would also expect the mammal diversity layers to be highly correlated with the climate-based global environmental stratification. It might be that mammal diversity is being selected because it better represents important broad-scale climate variability than a categorical stratification. The authors could test this by offering alternative continuous climate gradients to the analysis.

We looked at correlations between variables. We output a scatterplot matrix of a subset of variables which we manually selected, and added to that any variables with above 0.5 R^2 with any other variable in the model. This is included in Supplementary Notes.

Some association is indeed apparent between some predictors of interest, though not all is outright collinearity. As the reviewer suspected, there is a clear but idiosyncratic relationship between mammal diversity and GEnS. We have attempted to strike a balance between including a parsimonious set of predictors and selecting a broad enough set of predictors to, a priori, capture effects which may be present across multiple variables.

We had done some pruning of variables earlier in the modeling process. Previous versions included the entire set of bioclim variables in the model, and found that none of them showed up as very important and that replacing them all with GEnS did not change the model's predictive power much while considerably simplifying the model. Furthermore, GEnS was attractive for inclusion because it was generated through a principal components analysis of bioclim variables, and thus might capture meaningful variation spread across those variables.

For now, rather than picking one of, say, Evergreen Broadleaf Trees and Mammal Species Richness to remove, we think it better to include them both, and note the potential for the attenuation of effects. Future approaches may involve the generation of GEnS-like principal component analysis-based layers for variables thought to be part of the same causal pathway.

This comment did, however, prompt us to remove some of the variables which are not thought to be linked to EID events and had not been coming up as important in our models, namely Water (which was mainly present in included grid cells due to interpolation of ocean in coastal areas; not representing, say, rivers or inland water bodies, which *have* been found in other studies to be

associated with water-borne disease outbreaks; Yang et al., 2013), Snow and Ice (mostly present at the poles) and Barren.

Were interactions between variables, like between population density and population change or between livestock density and mammal diversity considered explicitly in the BRT modelling?

We probed for three-way interaction among variables. We have now included results of these tests, as means of interaction scores across replicate models in our Supplementary Results 1 and 2. We also internally explored interactions using 2-variable partial dependence plots, but we did not find that they offered much useful explanatory information above and beyond our current interpretations, especially as effects already show wide confidence intervals and displaying those in three dimensions is difficult — and we don't want to over-interpret.

I highly appreciate the efforts to correct for observation bias. However, the authors should give a more detailed treatment of the meaning and potential shortcomings of their observation bias measure. Is the open-access subset biased towards researchers in richer locations that can pay open-access fees? Or is the subset spatially biased towards more recent research effort, mismatching with the temporal range of the EID data? The authors should specify the temporal range of the publication effort data on page 22.

The model output submitted with these revisions includes an updated version of the reporting effort measure. In response to the comments of all the reviewers, we have included a detailed SI section on the methods used to create the reporting effort dataset. We are hosting the PubCrawler NLP code publicly on GitHub (URLs provided above) and plan to continue development of the tools. The PubMed Central Open Access Subset (PMCOAS) is large and extends back in time, but only recent articles are available in the structured XML format needed for processing, and the vast majority of the OAS is very recent (anecdotally, since the first iteration of the PubCrawler dataset, run about a year ago, to the version run during these revisions, the number of papers available has increased from ~800,000 to ~1,300,000. Hence, our reporting effort dataset could not be temporally matched in the same way as some of our covariates. This is now noted in Table 1.

Also, are the processes that lead to an EID event making it into the EID database, similar to the process that leads to an EID event being written up for publication. I assume most disease events are not initially notified by public health researchers but by public health authorities so are the suites of sites the same? To maximise the impact of this methodological improvement on the field, I would like to see the authors deal briefly with some of these issues and how they may impact their conclusions. Overall, I agree it is probably an advance over the prior metrics for observation bias referenced on page 22 and the impacts on the heatmap in Fig. 3 are interesting and important. It may be interesting to include in the supplementary methods part some brief mention of the key environmental drivers of publication effort, according to the BRT model that they used to smooth the raw data (pgs. 22-23).

The events in the EID event database were gathered from an extensive review of published scientific literature, which probably lags actual public health reporting somewhat (This is one of the reasons our database cuts off a few years ago, and has not been brought completely up to date). One of the additions made to the PubCrawler method is that we now process a subset of the PMCOAS based on search terms related to human infectious diseases (see SI), which hopefully makes the background weight and the presence dataset more comparable. We have included in the SI details of the BRT model used to smooth the data, including important variables, and a country-aggregated comparison with the previous study's bias metric. Despite these ongoing improvements, the revised reporting effort layer remained highly correlated with the original layer, so the results of the model remained similar.

Some claims are overstated on the basis of this particular analysis. I am not convinced from the explanation below that the authors can really distinguish between the "perturbation" of pathogen dynamics versus the exposure to a novel pathogen pool from this analysis. Isn't the amount of edge interface between natural and human modified habitats rather than the overall cover change the relevant landscape metric to disentangle these two?

"Inter-decadal pasture, cropland and human population change show less relative influence in explaining global patterns of EID occurrence. Their partial dependence plots, however, suggest increased EID event probability at both positive and negative values. This supports the hypothesis that land use change may cause "perturbation" of pathogen dynamics in a system, promoting cross-species transmission, as opposed to representing exposure to a novel pathogen pool due to encroachment on previously undisturbed habitat 10. Perturbation may increase emergence risk by impacting the number of infectious hosts, or altering pathogen prevalence as a function of habitat intactness by a range of mechanisms related to meta-community ecology 26, whereas we would not expect risk to also increase with decreasing pasture change under a novel pathogen pool explanation."

We've toned down this language. We agree with the reviewers that the BRT partial dependence plots are likely not interpretable at this level, especially given the wide confidence intervals.

Apart from the absence sampling, accuracy statistics and collinearity problems discussed above, the statistics are appropriate, including in the treatment and presentation of the uncertainty in the analysis. There is insufficient detail about the BRT for replication of the study. The authors should add the learning rate, tree complexity and bag rate settings of the BRT in the supplementary methods section.

We have expanded the detail on BRT as suggested. See also response to Reviewer 2.

Specific comments

Page 3. Please reiterate the dependent variable for readers unfamiliar with 4. **Done (see response to Reviewer 2).**

Page 11. Move line about " increase in zoonotic EID relative risk at low levels of mammal species richness" to the results. **We added this to the Results section, and left this phrase in the Discussion to**

contextualize the following sentences.

Page 12. Some elements of this discussion are quite speculative and not particularly well linked to the results of this particular analysis.

For example, this statement seems rather obvious: "Indeed, both positive and negative effects of wildlife richness on disease risk could be occurring, depending, for example, on geographic region, spatial scale, subgroups of diseases, host species composition, or a host of other factors." **This sentence has been removed.**

Page 12. "increase with decreasing pasture change under a novel pathogen pool explanation."

Shouldn't "decreasing pasture change"

The sections on model impacts and future work to be based on this framework are overlong and could be substantially reduced. **We have revised our language to stick closer to the data and to be more concise.**

Reviewers' comments:

Reviewer #1 (Remarks to the Author):

The ms. is somewhat improved and clarified from my perspective. I suppose that there are inherently subjective aspects to efforts like this, but one has to start somewhere with reasonable definition and assumptions, so I don't fault them for it.

I think it best not to quibble at this point: once the public sees the paper and the site, it will react appropriately with kudos and probably at least a few criticisms, which is as it should be.

Reviewer #2 (Remarks to the Author):

I would like to thank the authors for their thorough and considered response. Whilst I am still not 100% convinced of some of their claims, particularly around the PubMed scraper, they have provided ample supporting evidence and ways for readers to make their own judgement. Furthermore they have now caveated or discussed implications of some of the issues I previously had which is sufficient.

This article certainly improves upon the Jones et al. methodologies and I hope will remain a key article in discussing spatial aspects of disease emergence.

Reviewer #3 (Remarks to the Author):

Review of revised manuscript by Allen et al.

I was very satisfied with the way the authors addressed many of the original reviewer comments including those on collinearity, improved description of the BRT and reporting effort metrics, and toning down interpretation of predictor effects. However, I have some remaining methodological concerns in relation to the Boosted Regression Tree modelling which may impact the main conclusions about the key environmental drivers of EID events. I would recommend that these are addressed before acceptance.

Firstly, I am concerned that the predictive performance of the weighted model is so poor (indicated by an AUC of 0.65), particularly because the authors used a method of cross-validation, leave-one out cross-validation, that is not very robust to over-fitting and is a poor indicator of a model's ability to predict independent data. It is more usual in ecology and epidemiology to see BRT models validated by several fold cross-validation where up to 1/10th of the data is excluded from model fitting (Elith et al. 2008) rather than a single data point.

A more minor issue that affects the predictions of the model to new areas (rather than the overall accuracy) is that the tuning parameters of the model stated in lines 440-441 (tree.complexity = 3, learning.rate = 0.0035, and 441 n.trees = 35) contravene the rules of thumb established by Elith et al. (2008) that a learning rate for a given tree complexity should be selected that allows the BRT model to fit at least 1000 trees (n.trees). This is to minimise instability between-models in predictions to independent samples, caused by the stochastic component of model building, in which random subsets of data are selected for fitting each tree. Could the authors explain how their BRT model settings were tuned – what were their selection criteria?

For the reporting effort model, I am concerned that the data are likely over-dispersed (requiring a negative binomial) or zero-inflated (requiring a zero-inflated poisson or negbin model). Can the authors provide further statistics to show that the choice of a poisson model was appropriate here?

There seem to be some geographical anomalies in Fig. 3. That are not mentioned in the results or

discussion. For example, the south of England is predicted to be at high relative risk of zoonotic disease events.

Colour bar of Fig. 4 needs careful labelling before it can be interpreted.

Lines 159 to 162 about the inclusion of population density in both the reporting effort model and the EID model, and what this means for inference, is hard to read and needs rewriting.

"Population density was also included in the reporting effort model, but was not as strong a predictor (rel. inf. 3.6%) as urban land (rel. inf. 45.2%). It is possible that a positive association between EID events and overall reporting effort is the assumed baseline multiplicative effect of population density on events, modulated by the distribution of reporting effort, which appears to be strongly associated with urbanization — all of which are factored out by our weighted model."

Supplementary information

Check language here: "We associated them with JSON shapefiles and projected them to the study grid, then multiplied them by the population in each grid cell to bring their units in line with other variables in the model." Do you mean "divided them by the population in each grid cell"?

Supplementary Figures 3 and 4 require colour bars and clearer descriptions in the legend of the quantities being mapped.

Supp fig 5, needs reformatting with appropriate variable abbreviations so that the variable names do not overlap and can all be read. Should legend read "automatically excluded if $R^2 > 0.5$ " rather than "automatically included"? Also presumably the matrix shows a pearson's r coefficient rather than an R^2 .

Reviewers' comments:

We thank all Reviewers for their acknowledgment of our efforts to satisfy their previous comments on our work.

Reviewer #1 (Remarks to the Author):

The ms. is somewhat improved and clarified from my perspective. I suppose that there are inherently subjective aspects to efforts like this, but one has to start somewhere with reasonable definition and assumptions, so I don't fault them for it.

I think it best not to quibble at this point: once the public sees the paper and the site, it will react appropriately with kudos and probably at least a few criticisms, which is as it should be.

We appreciate this Reviewer's final comments and all their previous suggestions that greatly improved our manuscript.

Reviewer #2 (Remarks to the Author):

I would like to thank the authors for their thorough and considered response. Whilst I am still not 100% convinced of some of their claims, particularly around the PubMed scraper, they have provided ample supporting evidence and ways for readers to make their own judgement. Furthermore they have now caveated or discussed implications of some of the issues I previously had which is sufficient. This article certainly improves upon the Jones et al. methodologies and I hope will remain a key article in discussing spatial aspects of disease emergence.

We appreciate these positive comments and thank this reviewer for their earlier suggestions that improved our manuscript.

Reviewer #3 (Remarks to the Author):

Review of revised manuscript by Allen et al.

I was very satisfied with the way the authors addressed many of the original reviewer comments including those on collinearity, improved description of the BRT and reporting effort metrics, and toning down interpretation of predictor effects. However, I have some remaining methodological concerns in relation to the Boosted Regression Tree modelling which may impact the main conclusions about the key environmental drivers of EID events. I would recommend that these are addressed before acceptance.

We are pleased to hear that the Reviewer is satisfied with the way we addressed their previous comments. We address the other specific methodological issues in detail below.

Firstly, I am concerned that the predictive performance of the weighted model is so poor (indicated by an AUC of 0.65), particularly because the authors used a method of cross-validation, leave-one out cross-validation, that is not very robust to over-fitting and is a poor indicator of a model's ability to predict independent data. It is more usual in ecology and epidemiology to see BRT models validated by several fold cross-validation where up to 1/10th of the data is excluded from model fitting (Elith et al. 2008) rather than a single data point.

We appreciate the reviewer's concerns about adequately conveying predictive performance. We rewrote the sampling regime for our cross-validation model to use ten-fold cross-validation, per their suggestion. All cross-validation statistics included in the paper now reference the output of the new process, and the Results section describe it in more detail. This does not affect the general fitting process of the model, which — using the `gbm.step()` function from the `dismo` package — already iteratively used ten-fold cross-validation to select the appropriate number of trees to fit.

The AUC are slightly lower (0.61 for the weighted model, 0.78 compared to unweighted residuals and 0.77 for the unweighted model). This is expected, and likely partially due to the slightly more stringent nature of this cross-validation, and partially due to the fact that we are removing an additional 10% of the data from a model which is already data-constrained.

We posit that low-to-moderate AUC in our weighted model is such because the model is essentially fit and evaluated on the residuals of emergence relative to reporting effort, itself a strong predictor of where we observe events. We have continued our strategy of leading with this model, despite the higher AUC of the unweighted model, because we feel that this better portrays a clear picture of the limitations of available data and reporting, and we have attempted to constrain our interpretations to be appropriate to this.

A more minor issue that affects the predictions of the model to new areas (rather than the overall accuracy) is that the tuning parameters of the model stated in lines 440-441 (`tree.complexity = 3`, `learning.rate = 0.0035`, and `441 n.trees = 35`) contravene the rules of thumb established by Elith et al. (2008) that a learning rate for a given tree complexity should be selected that allows the BRT model to fit at least 1000 trees (`n.trees`). This is to minimise instability between-models in predictions to independent samples, caused by the stochastic component of model building, in which random subsets of data are selected for fitting each tree. Could the authors explain how their BRT model settings were tuned – what were their selection criteria?

We have taken on board all of these suggestions as follows: Firstly, we note that the parameters that the reviewer references are the parameters governing the `gbm.step()` function's stepwise fitting process. In this function, the `n.trees` parameter specifies the number of trees fit at the start of this process, and (unless otherwise specified by the user) the `step.size` parameter during model fitting, indicating the number of new trees added at each new iteration of the stepwise fitting process. We agree that this was not clear enough in the paper, especially considering that the `gbm()` function, from the `gbm` package, internally called by `gbm.step()`, uses `n.trees` to indicate the number of trees in the model. We have reworded this section to be more precise.

We tuned the models iteratively, following examples and methods laid out in Elith et al. (2008) "A working guide to boosted regression trees", Leathwick et al. 2006 "Variation in demersal fish species richness in the oceans surrounding New Zealand: an analysis using boosted regression trees", and others (including Elith & Hastie's documentation for the `dismo` package). Starting with the default

values (`tree.complexity = 1`, `learning.rate = 0.01`, `n.trees = 50`), we monitored metrics like number of final trees, holdout deviance explained, and the graphical trace of the gradient descent, increasing `tree.complexity` and reducing `learning.rate` and `n.trees` to a point that achieved consistent results, with models fitting successfully, with a satisfactory number of trees.

The individual BRTs that make up our final bootstrap model have a mean 1005 trees per model, which we now report in the paper. This approximately hits Elith et al. (2008)'s rule of thumb for a single model. Since our bootstrap output aggregates those all models, and includes them in our empirical confidence intervals, models with fewer trees are not interpreted individually and contribute to the spread of uncertainty we convey.

However, to ensure sure that this was not detrimentally impacting our models, we did fit and examine a version of the bootstrap model with a lower `learning.rate` parameter. The mean number of trees in the fitted GBMs was 1373. The raw output (before reweighting) was highly correlated with the fewer-trees version used in the paper ($R^2 = 0.9985$). We determined that this was small enough to be of little import to our interpretations, and went with the smaller, more computationally manageable versions of the models.

For the reporting effort model, I am concerned that the data are likely over-dispersed (requiring a negative binomial) or zero-inflated (requiring a zero-inflated poisson or negbin model). Can the authors provide further statistics to show that the choice of a poisson model was appropriate here?

The reviewer is correct in their supposition that the data do display clear over-dispersion, and excess zeroes. In the raw publications metric, the variance (4919.2) is much higher than the mean (7.5). This is likely due to the large number of grid cells globally with zero mentions found in the literature analyzed. If we were using a GLM framework, zero-inflated or negative binomial regression would be the appropriate way to deal with this.

We had explored GLM frameworks earlier on in our modeling process, including:

- Poisson GLM, which found all variables be significantly associated with the outcome, but was inappropriate for use because of overdispersion;
- Quasipoisson GLMs, which had large dispersion parameters and non-significant p-values;
- `glm.nb()` function from the `pscl` package, for a negative binary GLM, which failed to fit;
- Zero-inflated Poisson models using the `zeroinfl()` function from the `boot` package, which encountered problems fitting.

We posit that the fit problems are due to the high number of zeroes (approximately 78% of grid cells were zero) and high-valued outliers among grid cells, which do not conform to the distributional requirements the GLMs and their fitting algorithms.

We also attempted to remedy the GLM fit problems and potential overdispersion in other ways.

- In case the large number of zeroes was caused by a lack of GeoNames entities in the zero grid cells, we created a variable by summing the number of eligible GeoNames entities in all grid cells (using the same categories as were used to match locations in PubMed Central text). We included this as a coefficient and as an offset in the models fit. Including this as a coefficient and as an offset in the models did not solve the problems the GLM modules encountered

As stated elsewhere, we selected BRTs largely because they are able to return fairly accurate predictions — as long as they are predicting values which occur within the covariate space in which they were trained — and are robust to data which would be problematic or assumption-violating for other modeling methods, including sparse data and non-normally distributed data. The predominant R packages for boosted regression trees do not include variants of zero-inflated Poisson or negative binomial regression.

In this case, the Poisson BRT's flexibility also allows it to handle the data's overdispersion. BRTs aggregate models fit to subsections of covariate space, and this allows different values for covariates (and λ) in different subsections of covariate space. This permits the model to capture the higher variance that results from overdispersion/excess zeroes. The prediction layer from the Poisson BRT indeed exhibits similar overdispersion to the raw layer, with a mean of 7.9 and variance of 3232.2.

We have added the majority of this text to Supplementary Information describing the PubCrawler layer.

There seem to be some geographical anomalies in Fig. 3. That are not mentioned in the results or discussion. For example, the south of England is predicted to be at high relative risk of zoonotic disease events.

The reviewer is correct: the second panel in Figure 3 — the “bias factored out” version — shows population centers at elevated risk for EID events, and this falls in line with what we'd expect — as disease events in the human population, we do expect to see more of these occurring at areas of high population density. Our bias-factored-out map factors out total reporting effort, and factors back in population. This was described in the caption to Figure 4 (the which shows model's raw output between those two steps) and in the Results, but we have added parenthetical statements to Figure 3's caption to explain the process there, too, and refer the reader to Figure 4. We added a sentence to the Results text to note that “areas of high population, such as cities in Europe and the United States, remain among areas at the high end of the risk index”. We also added a note to the discussion of the model's limitations stating that because of the model's uncertainty, we refrain from making specific i.e. city-by-city interpretations of the risk index. We thank the reviewer for noting this.

Colour bar of Fig. 4 needs careful labelling before it can be interpreted.

We have added a numeric scale to Figure 4, which is consistent with the scale in the partial dependence plots. Per the Figure 2 legend: “Our sampling regime fixes the outcome, which indexes EID event risk, between 0 and 1, with a mean at 0.5”. In terms of the grid, the output shows the probability of that grid cell containing an EID event — assuming that you're drawing grid cells at the rate of reporting effort. However, we left the Figure 3 captions, which have been reweighted with publication effort and population, with legends reading “High” and “Low”, because their numerical values are small and not meaningful in absolute terms (i.e. they sum to 1 across the grid cells).

Lines 159 to 162 about the inclusion of population density in both the reporting effort model and the EID model, and what this means for inference, is hard to read and needs rewriting.

“Population density was also included in the reporting effort model, but was not as strong a predictor (rel. inf. 3.6%) as urban land (rel. inf. 45.2%). It is possible that a positive association between EID events and overall reporting effort is the assumed baseline multiplicative effect of population density on events, modulated by the distribution of reporting effort, which appears to be strongly associated with urbanization — all of which are factored out by our weighted model.”

We thank the reviewer for pointing this out. We have taken another shot at clarifying the section, and also refer the reader to our more detailed discussion of our treatment of the reporting effort problem in Methods. The new section reads as follows:

“Theoretically, population has a baseline multiplicative effect on human disease events²⁸ — of which EID events are a subclass — and their detection is modulated by reporting effort. Reporting effort appears to be associated with urbanization, but reporting effort and urbanization are also both products of human population. We did not attempt to fully disentangle these factors, instead using our measure of reporting effort to present a map of emerging infectious disease hotspots with bias “factored out” (described below in Materials & Methods).”

Supplementary information

Check language here: “We associated them with JSON shapefiles and projected them to the study grid, then multiplied them by the population in each grid cell to bring their units in line with other variables in the model.” Do you mean “divided them by the population in each grid cell”?

Our original language was perhaps unclear. We meant that the *per capita* variables were weighted by population per grid cell — i.e. “GDP / capita” for the USA, read to the study grid, was weighted by the population distribution of the USA. We have updated the text to attempt to make this point more clearly.

Our outcome variable is measured at the grid cell level. Although GDP per capita in the USA, for example, is *reported* at country level in our source data, we know it isn’t distributed uniformly across

the country. Multiplying by the population in each grid cell yields a rough estimate of GDP per grid cell (literally *GDP / capita * capita*).

Supplementary Figures 3 and 4 require colour bars and clearer descriptions in the legend of the quantities being mapped.

We have added labelled color bars, and additional explanatory text to the captions.

Supp fig 5, needs reformatting with appropriate variable abbreviations so that the variable names do not overlap and can all be read. Should legend read “automatically excluded if $R^2 > 0.5$ ” rather than “automatically included”? Also presumably the matrix shows a pearson’s r coefficient rather than an R^2 .

We have attempted to make the longer variable names more readable in this figure — they now state the full variable name, but contain smaller text to prevent overlapping. We corrected the caption of the figure to state the correct statistic, and clarify the way that correlated pairs were included in the plot matrix.

REVIEWERS' COMMENTS:

Reviewer #3 (Remarks to the Author):

The authors have done a thorough job of addressing my technical concerns on the methodology and presenting this in a clear way in the manuscript. I am very happy that this paper can now be published in its current form and will make an excellent contribution to the field.